# On Computation and Generalization of Generative Adversarial Networks under Spectrum Control

**Haoming Jiang, Zhehui Chen, Minshuo Chen & Tuo Zhao** [*]
School of Industrial and Systems Engineering
Georgia Institute of Technology
Atlanta, GA 30318, USA
{jianghm, zhchen, mchen393, tourzhao}@gatech.edu

**Feng Liu & Dingding Wang**
Department of Computer & Electrical Engineering and Computer Science
Florida Atlantic University
Boca Raton, FL 33431, USA
{fliu2016, wangd}@fau.edu

## Abstract

Generative Adversarial Networks (GANs), though powerful, is hard to train. Several recent works (Brock et al., 2016; Miyato et al., 2018) suggest that controlling the spectra of weight matrices in the discriminator can significantly improve the training of GANs. Motivated by their discovery, we propose a new framework for training GANs, which allows more flexible spectrum control (e.g., making the weight matrices of the discriminator have slow singular value decays). Specifically, we propose a new reparameterization approach for the weight matrices of the discriminator in GANs, which allows us to directly manipulate the spectra of the weight matrices through various regularizers and constraints, without intensively computing singular value decompositions. Theoretically, we further show that the spectrum control improves the generalization ability of GANs. Our experiments on CIFAR-10, STL-10, and ImgaeNet datasets confirm that compared to other methods, our proposed method is capable of generating images with competitive quality by utilizing spectral normalization and encouraging the slow singular value decay.

## 1 Introduction

Many efforts have been recently devoted to studying Generative Adversarial Networks (GANs, Goodfellow et al. (2014)). GANs provide a general unsupervised framework to learn a generative model from unlabeled real data. Successful applications of GANs include many unsupervised learning tasks, such as image generation, dialogue generation, and image inpainting (Abadi & Andersen, 2016; Goodfellow, 2016; Ho & Ermon, 2016; Li et al., 2017; Yu et al., 2018). Different from other unsupervised learning methods, which directly maximize the likelihood of deep generative models (e.g., Variational Auto-encoder, Nonlinear ICA, and Restricted Boltzmann Machine), GANs introduce a competition between two neural networks. Specifically, one neural network serves as the generator that yields artificial samples, and the other serves as the discriminator that distinguishes the artificial samples from the real data.

Mathematically, GANs can be formulated as the following min-max optimization problem:

$$\min_{\theta} \max_{\mathcal{W}} f(\theta, \mathcal{W}) := \frac{1}{n} \sum_{i=1}^{n} \phi\left(\mathcal{A}(D_{\mathcal{W}}(x_i))\right) + \mathbb{E}_{x \sim \mathcal{D}_{G_\theta}}[\phi\left(1 - \mathcal{A}(D_{\mathcal{W}}(x))\right)], \quad (1)$$

where $\{x_i\}_{i=1}^{n}$ are $n$ real data points, $G_\theta$ denotes the generative deep neural network parameterized by $\theta$, $D_{\mathcal{W}}$ denotes the discriminative neural network parameterized by $\mathcal{W}$, $\mathcal{D}_{G_\theta}$ denotes the distribution

---

[*]Tuo Zhao is the corresponding author.

generated by $G_\theta$, $\phi(\cdot) : [0, 1] \to \mathbb{R}$ is a properly chosen monotone function, and $\mathcal{A}(\cdot)$ denotes a monotone function related to the function $\phi(\cdot)$. There have been many options for $\phi(\cdot)$ and $\mathcal{A}(\cdot)$ in existing literature. For example, the original GAN proposed in Goodfellow et al. (2014) chooses $\phi(x) = \log(x)$, $\mathcal{A} = \frac{1}{1+\exp(-x)}$; Arjovsky et al. (2017) use $\phi(x) = x$, $\mathcal{A}(x) = x$, and (1) becomes the Wasserstein GAN. Min-max problem (1) has a natural interpretation: The minimization problem aims to find a discriminator $D_\mathcal{W}$, which can distinguish between the real data and the artificial samples generated by $G_\theta$, while the maximization problem aims to find a generator $G_\theta$, which can fool the discriminator $D_\mathcal{W}$. From the perspective of game theory, the generator and discriminator are essentially two players competing with each other and eventually achieving some equilibrium.

From an optimization perspective, problem (1) is a nonconvex-nonconcave min-max problem, that is, $f(\theta, \mathcal{W})$ is nonconvex in $\theta$ given a fixed $\mathcal{W}$ and nonconcave in $\mathcal{W}$ given a fixed $\theta$. Unlike convex-concave min-max problems, which have been well studied in existing optimization literature, there is very limited understanding of general nonconvex-nonconcave min-max problems. Thus, most of existing algorithms for training GANs are heuristics. Although some theoretical guarantees have been established for a few algorithms, they all require very strong assumptions, which are not satisfied in practice (Heusel et al., 2017).

Despite of the lack of theoretical justifications, significant progress has been made in empirical studies of training GANs. Numerous empirical evidence has suggested several approaches for stabilizing the training of the discriminator, which can eventually improve the training of the generator. For example, Goodfellow et al. (2014) adopt a simple algorithmic trick that updates $\mathcal{W}$ for multiple iterations after updating $\theta$ for one iteration, i.e., training the discriminator more frequently than the generator. Besides, Xiang & Li (2017) suggest that the weight normalization approach proposed in Salimans & Kingma (2016) can also stabilize the training of the discriminator. More recently, Miyato et al. (2018) propose a spectral normalization approach to control the spectral norm of the weight matrix in each layer. Specifically, in each forward step, they normalize the weight matrix by the approximation of its spectral norm, which is obtained by the one-step power method. They further show that spectral normalization essentially controls the Lipschitz constant of the discriminator with respect to the input. Compared to other methods for controlling the Lipschitz constant of the discriminator, e.g., gradient penalty (Gulrajani et al., 2017; Gao et al., 2017), the experiments in Miyato et al. (2018) show that the spectral normalization approach achieves better performance with fairly low computational cost. Moreover, Miyato et al. (2018) show that spectral normalization suffers less from the mode collapse, that is, the generator outputs only over a fairly small support. Such a phenomenon, though not well understood, suggests that the spectral normalization will balance the discrimination and representation well.

Besides the aforementioned algorithmic tricks and normalization approaches, regularization can also stabilize the training of the discriminator (Brock et al., 2016; Roth et al., 2017; Nagarajan & Kolter, 2017; Liu et al., 2018). For instance, orthogonal regularization, proposed by Brock et al. (2016), forces the columns of weight matrices in the discriminator to be orthonormal by augmenting the objective function $f(\theta, \mathcal{W})$ with $\lambda \sum_{i=1}^{L} \|W_i^\top W_i - I\|_{\mathrm{F}}$, where $\lambda > 0$ is the regularization parameter, $W_i$ denotes the weight matrix of the $i$-th layer in the discriminator, $I$ denotes the identity matrix, and $L$ is the depth of the discriminator. The experimental results in Brock et al. (2016) show that the orthogonal regularization improves the performance and generalization ability of GANs. However, the empirical evidence in Miyato et al. (2018) shows that the orthogonal regularization is still less competitive than the spectral normalization approach. One possible explanation is that the orthogonal normalization, forcing all non-zero singular values to be 1, is more restrictive than the spectral normalization, which only forces the largest singular value of each weight matrix to be 1.

Motivated by the spectral normalization, we propose a novel training framework, which provides more flexible and precise control over the spectra of weight matrices in the discriminator. Specifically, we reparameterize each weight matrix $W_i \in \mathbb{R}^{d_i \times d_{i+1}}$ as $W_i = U_i E_i V_i^\top$, where $U_i$ and $V_i$ are required to have orthonormal columns, $E_i = \mathrm{diag}(e_1^i, ..., e_{r_i}^i)$ denotes a diagonal matrix with $r_i = \min(d_i, d_{i+1})$, and $e_1^i \geq \cdots \geq e_{r_i}^i \geq 0$ are singular values of $W_i$. With such a reparameterization, an $L$-layer discriminator becomes

$$D(x; \mathcal{U}, \mathcal{E}, \mathcal{V}) = U_L E_L V_L^\top \sigma_{L-1}(\cdots \sigma_1(U_1 E_1 V_1^\top x) \cdots),$$

where $\sigma_i(\cdot)$ is the entry-wise activation operator of the $i$-th layer, $\mathcal{U} := \{U_1, ..., U_L\}$, $\mathcal{E} := \{E_1, ..., E_L\}$, and $\mathcal{V} := \{V_1, ..., V_L\}$ denote the parameters of the discriminator $D$, and $x$ denotes the input vector. This reparameterization allows us to control the spectra of the original weight matrix $W_i$

by manipulating $E_i$. For example, we can rescale $E_i$ by its largest diagonal element, which essentially is the spectral normalization. Besides, we can also manipulate the diagonal entries of $E_i$ to control the decays in singular values (e.g., fast or slow decays). Recall that our reparameterization requires $U_i$ and $V_i$ to have orthonormal columns. This requirement can be achieved by several methods in the existing literature, such as the stiefel manifold gradient method. However, Huang et al. (2017) show that the stochastic stiefel manifold gradient method is unstable. Moreover, other methods, such as cayley transformation and householder transformation, suffer from several disadvantages: (I). High computational cost[1]; (II). Sophisticated implementation (Shepard et al., 2015). Different from the methods mentioned above, our framework applies the orthogonal regularization to all $U_i$'s and $V_i$'s. Such a regularization suffices to guarantee the approximate orthogonality of $U_i$'s and $V_i$'s in practice, which is supported by our experiments. Moreover, our experimental results on CIFAR-10, STL-10 and ImageNet datasets show that our proposed method achieves competitive performance on CIFAR-10 and better performance than the spectral normalization and other competing approaches on STL-10 and ImageNet. Besides the empirical studies, we provide theoretical analysis, which characterizes how the spectrum control benefits the generalization ability of GANs. Specifically, denote $\mu$ as the underlying data distribution and $\nu_n$ as the distribution given by the well trained generator. We establish a generalization bound under spectrum control as follows (informal):

$$d_{\mathcal{F},\phi}(\mu,\nu_n) \leq \inf_{\nu \in \mathcal{D}_G} d_{\mathcal{F},\phi}(\mu,\nu) + \widetilde{O}\left(\sqrt{\frac{d^2 L}{n}}\right),$$

where $d = \max\{d_1, \ldots, d_L\}$, $d_{\mathcal{F},\phi}(\cdot, \cdot)$ is the $\mathcal{F}$-distance, and $\mathcal{D}_G$ denotes the class of distributions generated by generators. Compared to the results in Zhang et al. (2017), our result improves the generalization bound up to an exponential factor of the depth of the discriminator. More details will be discussed in Section 3.

The rest of the paper is organized as follows: Section 2 introduces our proposed training framework in detail; Section 3 presents the generalization bound for GANs under spectrum control; Section 4 presents numerical experiments on CIFAR-10, STL-10, and ImageNet datasets.

**Notations**: Given an integer $d > 0$, we denote $[d] = \{1, 2, ..., d\}$. Given a vector $v \in \mathbb{R}^d$, we denote $\|v\|_2^2 = \sum_{i=1}^d |v_i|^2$ as its Euclidean norm. Given a matrix $M \in \mathbb{R}^{m \times n}$, we denote the spectral norm by $\|M\|_2$ as the largest singular value of $M$. We adopt the standard $O(\cdot)$ notation, which is defined as $f(x) = O(g(x))$ as $x \to \infty$, if and only if there exists $M > 0$ and $x_0$, such that $|f(x)| \leq Mg(x)$ for $x \geq x_0$. We use $\widetilde{O}(\cdot)$ to denote $O(\cdot)$ with hidden logarithmic factors.

## 2 METHODOLOGY

We present a new framework for flexibly controlling the spectra of weight matrices. We first consider an $L$-layer discriminator $D$ as follows:

$$D(x; \mathcal{W}) = W_L \sigma_{L-1}(W_{L-1} \cdots \sigma_1(W_1 x) \cdots), \tag{2}$$

where $\sigma_i(\cdot)$ denotes the entry-wise activation operator of the $i$-th layer, $W_i \in \mathbb{R}^{d_{i+1} \times d_i}$ denotes the weight matrix of the $i$-th layer, $x \in \mathbb{R}^{d_1}$ denotes the input feature, $\mathcal{W} := \{W_1, ..., W_L\}$ denotes the parameters of the discriminator $D$, and $d_{L+1} = 1$.

### 2.1 SVD REPARAMETERIZATION

Our framework directly applies an SVD reparameterization to each weight matrix $W_i$ in the discriminator $D$, i.e., $W_i = U_i E_i V_i^\top$, where $r_i = \min(d_i, d_{i+1})$, $U_i \in \mathbb{R}^{d_{i+1} \times r_i}$ and $V_i \in \mathbb{R}^{d_i \times r_i}$ denote two matrices with orthonormal columns, $E_i = \mathrm{diag}(e_1^i, \cdots, e_{r_i}^i)$ denotes a diagonal matrix, and $e_1^i \geq \cdots \geq e_{r_i}^i \geq 0$ are the singular values of $W_i$. The discriminator can be rewritten as follows:

$$D(x; \mathcal{U}, \mathcal{E}, \mathcal{V}) = U_L E_L V_L^\top \sigma_{L-1}(U_{L-1} E_{L-1} V_{L-1} \cdots \sigma_1(U_1 E_1 V_1^\top x) \cdots)), \tag{3}$$

where $\mathcal{U} := \{U_1, ..., U_L\}$, $\mathcal{E} := \{E_1, ..., E_L\}$, and $\mathcal{V} := \{V_1, ..., V_L\}$[2] denote the parameters of the discriminator $D$. Throughout the rest of the paper, if not clear specified, we denote $D(x; \mathcal{U}, \mathcal{E}, \mathcal{V})$

---

[1] Without a sparse matrix implementation, these methods are highly unscalable and inefficient (not supported by the existing deep learning libraries such as TensorFlow and PyTorch in GPU).

[2] $W_L$ essentially is a vector. To be consistent, we still use $U_L E_L V_L^\top$ to reparametrize $W_L$. Actually, it is not necessary. We can directly control the norm of $W_L$ in practice.

by $D(x)$ for notational simplicity. The motivation behind this reparameterization is to control the singular values of each weight matrix $W_i$ by explicitly manipulating $E_i$. We then consider a new min-max problem as follows:

$$\min_{\theta} \max_{\mathcal{E},\mathcal{U},\mathcal{V}} \bigg\{ \underbrace{\frac{1}{n}\sum_{i=1}^{n} \phi\left(\mathcal{A}(D(x_i))\right) + \mathbb{E}_{x\sim\mathcal{D}_{G_\theta}}[\phi\left(1-\mathcal{A}(D(x))\right)]}_{f(\theta,\mathcal{E},\mathcal{U},\mathcal{V})} -\gamma\mathcal{R}(\mathcal{E})\bigg\},$$

$$\text{subject to } \mathcal{E}\in\Omega,\ \ U_i^\top U_i = I_i,\ \text{ and }\ V_i^\top V_i = I_i\ \ \forall i\in[L], \tag{4}$$

where $I_i$ denotes the identity matrix of size $r_i$, $\mathcal{R}(\mathcal{E})$ is the regularizer with a regularization parameter $\gamma > 0$, and $\Omega$ denotes a feasible set. By choosing different $\Omega$ and $\mathcal{R}(\mathcal{E})$, (4) can control the spectrum of the weight matrix $W_i$ flexibly. For example, if we take the feasible set $\Omega = \{\mathcal{E} : e_j^i = 1 \ \forall e_j^i \in E_i, E_i \in \mathcal{E}\}$ and $\mathcal{R}(\mathcal{E}) = 0$, then our method essentially is the **orthogonal regularization**. We will discuss some options of $\Omega$ and $\mathcal{R}(\mathcal{E})$ later in detail.

As mentioned earlier, the orthogonal constraints in (4) suffer from the high computational cost and sophisticated implementation. To address these drawbacks, we directly apply the orthogonal regularization to all $U_i$'s and $V_i$'s. Therefore, problem (4) becomes

$$\min_{\theta} \max_{\mathcal{E},\mathcal{U},\mathcal{V}} f(\theta,\mathcal{E},\mathcal{U},\mathcal{V}) - \lambda\sum_{i=1}^{L}(\|U_i^\top U_i - I_i\|_{\mathrm{F}}^2 + \|V_i^\top V_i - I_i\|_{\mathrm{F}}^2) - \gamma\mathcal{R}(\mathcal{E}),\ \text{s.t. } \mathcal{E}\in\Omega, \tag{5}$$

where $\lambda > 0$ is a regularization parameter. A relative large $\lambda$ (e.g., $\lambda = 1$), ensures the orthogonality of $U_i$ and $V_i$. See more details in Section 4.1. Moreover, (5) can be efficiently solved by stochastic gradient algorithms. Projection may be needed to handle the constraint $\Omega$. See more details later.

## 2.2 SPECTRUM CONTROL

We provide a few options of $\Omega$ and $\mathcal{R}(\mathcal{E})$ for controlling the spectra of weight matrices in the discriminator, which is motivated by Miyato et al. (2018). Miyato et al. (2018) have shown that for an $L$-layer discriminator $D$, we have:

$$|D(x) - D(y)| \leq \|W_L\|_2\left(\prod_{i=1}^{L-1}\|W_i\|_2\cdot\rho_i\right)\|x-y\|_2 = e_1^L\left(\prod_{i=1}^{L-1}e_1^i\cdot\rho_i\right)\|x-y\|_2, \tag{6}$$

where $\rho_i$ is the Lipschitz constant of $\sigma_i(\cdot)$. The last equation holds for our proposed reparameterization. For commonly used activation operators, such as the sigmoid, ReLU, and leak-ReLU functions, $\rho_i \leq 1$. Therefore, $\prod_{i=1}^{L} e_1^i$ is essentially an upper bound for the Lipschitz constant, which can be controlled by our proposed $\Omega$ and $\mathcal{R}(\mathcal{E})$. Note that $W_L$ is a vector with only one singular value. For simplicity, we set $e_1^L = 1$ in the following analysis.

### 2.2.1 FLEXIBLE SPECTRAL CONTROL

Comparing to the orthogonal regularization, Miyato et al. (2018) suggest that we should allow more flexibility by using spectral normalization, which only bounds the largest singular value. They implement spectral normalization by **one-step power iteration**.

• **Spectrum Normalization:** We can also easily implement spectral normalization under our SVD reparameterization framework. Specifically, the spectral normalization rescales the weight matrix $E_i$ by its spectral norm $e_1^i$, which is equivalent to solving the following problem:

$$\min_{\theta} \max_{\mathcal{Q}(\mathcal{E}),\mathcal{U},\mathcal{V}} f(\theta,\mathcal{Q}(\mathcal{E}),\mathcal{U},\mathcal{V}) - \lambda\sum_{i=1}^{L}(\|U_i^\top U_i - I_i\|_{\mathrm{F}}^2 + \|V_i^\top V_i - I_i\|_{\mathrm{F}}^2),$$

where $\mathcal{Q} := \{\frac{E_1}{e_1^1}, \ldots, \frac{E_L}{e_1^L}\}$.

• **Spectrum Constraint:** Note that the spectral normalization essentially reparameterize the Lipschitz constraint $\Omega$:

$$\Omega = \left\{\mathcal{E} : 0 \leq e_1^i \leq 1\ \forall i\in[L]\right\}. \tag{7}$$

This essentially controls $\prod_{i=1}^{L} e_1^i$ by forcing each $e_1^i \leq 1$. Instead of spectral normalization, we consider directly solving the problem with the Lipschitz constraint. To maintain the feasibility of $\mathcal{E}$,

we only need a simple projection for each $E_i$ in the back propagation, which can be implemented by a simple entry-wise clipping operator defined as

$$g(t) := 0 \cdot \mathbb{1}_{\{t \leq 0\}} + t \cdot \mathbb{1}_{\{0 < t < 1\}} + 1 \cdot \mathbb{1}_{\{t \geq 1\}}, \tag{8}$$

where $\mathbb{1}_{t \in A} = 1$ if $t \in A$, and 0 otherwise.

These two methods are essentially solving the same problem, but in different formulations. Therefore, different algorithms are adopted. Due to the nonconvex-nonconcave structure of (5), different solutions are obtained.

• **Lipschitz Regularizer:** We can also directly penalize $\prod_{i=1}^{L} e_1^i$ to control the Lipschitz constant of the discriminator $D$. Specifically, we define the Lipschitz regularizer as:

$$\mathcal{R}(\mathcal{E}) := \max \left( \log \prod_{i=1}^{L} e_1^i, 0 \right) = \max \left( \sum_{i=1}^{L} \log e_1^i, 0 \right).$$

Compared to the spectral constraint, which enforces all $e_1^i \leq 1$, the Lipschitz regularizer is more flexible since it allows $e_1^i > 1$ for some $i \in [L]$.

### 2.2.2 SLOW SINGULAR VALUE DECAY

Miyato et al. (2018) owe their empirical success of training SN-GAN to controlling the spectral norm while allowing flexibility. This perspective, however, is not very concrete. As we know, orthogonal regularization and spectral normalization with SVD can both control the spectral norm. Their empirical performance is actually worse than SN-GAN. For example, on the STL-10 dataset, SN-GAN achieves an inception score of 8.83, while singular value truncation only achieves 8.69 and orthogonal regularization achieves 8.77.

The reason behind is that SN-GAN implements the spectral normalization via one-step power iteration. This procedure consistently underestimates spectral norms of weight matrices. Consequently, in addition to controlling the spectral norms, the spectral normalization in SN-GAN affects the whole spectrum of the weight matrix (encourages slow singular value decay as in Figure 1), which we refer to as "flexibility". Encouraging slow decay is essentially encouraging the network to capture as many features as possible while allowing correlation between neurons. Built upon these empirical observations, we conjecture that controlling the whole spectrum better improves the performance of GANs, which is further corroborated by our numerical experiments (Section 4).

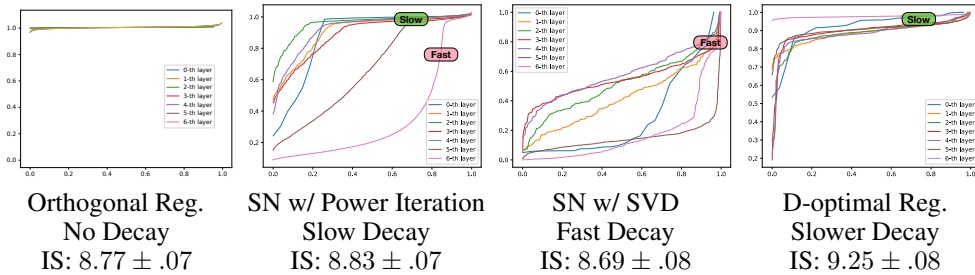

| Orthogonal Reg. | SN w/ Power Iteration | SN w/ SVD | D-optimal Reg. |
|---|---|---|---|
| No Decay | Slow Decay | Fast Decay | Slower Decay |
| IS: $8.77 \pm .07$ | IS: $8.83 \pm .07$ | IS: $8.69 \pm .08$ | IS: $9.25 \pm .08$ |

Figure 1: An illustration of smooth singular value decays with different methods. The vertical axis denotes the value and the horizontal axis denotes the normalized rank. The inception scores on STL-10 are also reported.

• **D-Optimal Regularizer:** We propose the $D$-Optimal Regularizer as follows:

$$\mathcal{R}(\mathcal{E}) = \frac{1}{2} \sum_{i=1}^{L-1} \log(|(E_i^\top E_i)^{-1}|) = -\sum_{i=1}^{L-1} \log \left( \prod_{k=1}^{r_i} e_k^i \right), \tag{9}$$

which is motivated by $D$-optimal design. $D$-optimal design (Wu & Hamada, 2011) is a popular principle in experimental design, where people aim to estimate parameters of statistical models with a minimum number of experiments. Specifically, $D$-optimal design maximizes the determinant of Fisher information matrix while allowing correlation between features in experiments. Existing literature has shown the superiority of $D$-optimal design to the orthogonal (uncorrelated) design on nonlinear model estimation (Yang et al., 2013; Li & Majumdar, 2009; Mentre et al., 1997).

Analogously, our proposed $D$-Optimal Regularizer essentially maximizes the log Gram determinant of the weight matrix,

$$\frac{1}{2}\sum_{i=1}^{L-1}\log(|(W_i^\top W_i)^{-1}|) \approx \frac{1}{2}\sum_{i=1}^{L-1}\log(|(E_i^\top E_i)^{-1}|).$$

The approximation holds due to the SVD reparameterization $W_i = U_i E_i V_i$, with $U_i, V_i$ approximately orthogonal. Moreover, note that the derivative of $\log(t)$ is $\frac{1}{t}$, a monotone decreasing function. Then $\log(t)$ has a significant impact when $t$ is small. Thus, $D$-optimal regularizer $\mathcal{R}(\mathcal{E})$ encourages a slow singular value decay.

• **Divergence Regularizer:** We propose a divergence regularization to precisely control the slow decay as shown in Figure 1. To mimic such a decay, we consider a reference distribution, $y = 1 - \min(|z|, 1)$, where $z \sim N(0, a^2)$. Figure 2 shows the decays of 10K order statistics sampled from $y$. We then denote the density function of $y$ as $p$ and the probability mass function of a uniform discrete distribution over $\{e_j^i\}_{j=1}^{r_i-1}$ as $Q_i(e_j^i) = \frac{1}{r_i-1}\forall j \in [r_i - 1]$. Note that the K-L divergence between a discrete distribution and a continuous distribution is $\infty$. To address this issue, we discretize $p$. Specifically, given $\{e_j^i\}_{j=1}^{r_i}$, we construct a discrete distribution over $\{e_j^i\}_{j=1}^{r_i-1}$ with a probability mass function $P_i^*$, defined as follows:

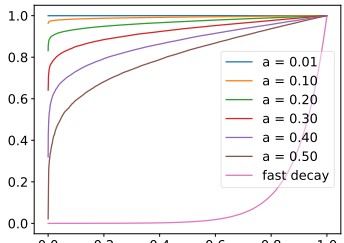

$$P_i^*(e_j^i) = \frac{p(e_j^i)(e_{j+1}^i - e_j^i)}{\sum_{k=1}^{r_i-1} p(e_k^i)(e_{k+1}^i - e_k^i)} \quad \forall j \in [r_i - 1].$$

Figure 2: The plot of normalized ranks versus values of 10K order statistics sampled from reference distributions. The vertical axis denotes the value; the horizontal axis denotes the normalized rank.

Ignoring the normalization term in the denominator, we then define the regularizer as follows:

$$\mathcal{R}(\mathcal{E}) = \sum_{i=1}^{L-1}\frac{1}{r_i-1}\sum_{k=1}^{r_i-1}\log\left[\frac{(r_i-1)^{-1}}{(e_{k+1}^i - e_k^i)p(e_k^i)}\right].$$

Note that the divergence regularizer requires the singular values in the interval $[0, 1]$ and $D$-optimal regularizer cannot control the Lipschitz constant of the discriminator $D$. Therefore, we incorporate the divergence regularizer with the spectrum constraint and combine the $D$-optimal regularizer with the spectral normalization to bound the Lipschitz constant. Our experimental results show that both combinations improve the training of GANs on CIFAR10 and STL-10 datasets.

## 3 THEORY

We show how the spectrum control benefits the generalization of GANs. Before proceed, we define $\mathcal{F}$-distance as follows.

**Definition 1** ($\mathcal{F}$-distance). *Let $\mathcal{F}$ be a class of functions from $\mathbb{R}^d$ to $[0, 1]$ such that if $f \in \mathcal{F}$, $1 - f \in \mathcal{F}$. Let $\phi$ be a concave function. Then given two distributions $\mu$ and $\nu$ supported on $\mathbb{R}^d$, the $\mathcal{F}$-distance $d_{\mathcal{F},\phi}(\mu, \nu)$ with respect to $\phi$ is defined as*

$$d_{\mathcal{F},\phi}(\mu, \nu) = \sup_{f \in \mathcal{F}} \mathbb{E}_{x \sim \mu}[\phi(f(x))] + \mathbb{E}_{x \sim \nu}[\phi(1 - f(x))] - 2\phi(1/2).$$

Note that $\mathcal{F}$-distance unifies Jensen-Shannon distance, Wasserstein distance and neural distance as proposed in Arora et al. (2017). For example, when taking $\phi(x) = x$ and $\mathcal{F} = \{$all 1-Lipschitz functions from $\mathbb{R}^d$ to $[0, 1]\}$, the $\mathcal{F}$-distance is the Wasserstein distance. Recall that by (1), the training of GANs is essentially minimizing the $\mathcal{F}$-distance with $\mathcal{F}$ being the collection of composite functions $\mathcal{A}(D(\cdot))$, where $D(\cdot)$ is the $L$-layer discriminator network defined by (2). To establish the generalization bound, we impose the following assumption.

**Assumption 1.** *The activation operator $\sigma_i$ is 1-Lipschitz with $\sigma_i(0) = 0$ for any $i \in [L-1]$. $\mathcal{A}$ is 1-Lipschitz such that if $\mathcal{A}(D(\cdot)) \in \mathcal{F}$, $1 - \mathcal{A}(D(\cdot)) \in \mathcal{F}$. $\phi$ is $\rho_\phi$-Lipschitz. The spectral norms of weight matrices are bounded respectively, i.e., $\|W_i\|_2 \le B_{W_i}$ for any $i \in [L]$.*

Note that commonly used functions $\mathcal{A}$, such as the sigmoid function, satisfy the assumption. We denote by $\mu$ the underlying data distribution, and by $\widehat{\mu}_n$ the empirical data distribution. We further denote $\nu_n$ as the distribution given by the generator that minimizes the loss (1) up to accuracy $\epsilon$, i.e.,

$$d_{\mathcal{F},\phi}(\widehat{\mu}_n, \nu_n) \leq \inf_{\nu \in \mathcal{D}_G} d_{\mathcal{F},\phi}(\widehat{\mu}_n, \nu) + \epsilon,$$

where $\mathcal{D}_G$ is the class of distributions generated by generators. Then we give the generalization bound based on the PAC-learning framework as follows.

**Theorem 2.** *Under Assumption 1, assume that the input data $x_i \in \mathbb{R}^{d_1}$ is bounded, i.e., $\|x_i\|_2 \leq B_x$ for $i \in [n]$. Then given activation operators $\sigma_1, \ldots, \sigma_{L-1}$, $\mathcal{A}$, and $\phi$, with probability at least $1 - \delta$ over the joint distribution of $x_1, \ldots, x_n$, we have*

$$d_{\mathcal{F},\phi}(\mu, \nu_n) \leq \inf_{\nu \in \mathcal{D}_G} d_{\mathcal{F},\phi}(\mu, \nu) + O\left( \frac{\rho_\phi \beta \sqrt{d^2 L \log\left(\sqrt{dn}L\beta\right)}}{\sqrt{n}} + \rho_\phi \beta \sqrt{\frac{\log \frac{1}{\delta}}{n}} \right) + \epsilon,$$

*where $\beta = B_x \prod_{i=1}^{L} B_{W_i}$ and $d = \max(d_1, \ldots, d_L)$.*

The detailed proof is provided in Appendix A.1. By constraining each $B_{W_i} = 1$, the generalization bound is reduced to of the order $\widetilde{O}\left(\sqrt{d^2 L/n}\right)$, which is polynomial in $d$ and $L$. On the contrary, without such spectrum constraints, the bound can be exponentially dependent on $L$. For example, if $B_{W_i} \geq 1 + r$ with some constant $r > 0$ for any $i = 1, \ldots, L$, we have $\beta \geq B_x(1 + r)^L$, which implies that GANs cannot generalize with polynomial number of samples.

**Remark 3.** *Empirical Rademacher complexity (ERC) is adopted to derive our generalization bound, which is of the order $\widetilde{O}\left(\beta\sqrt{d^2 L/n}\right)$. Directly applying the ERC based generalization bound in Bartlett et al. (2017) yields a bound of the order $\widetilde{O}\left(\beta\sqrt{d^2 L^3/n}\right)$. Our bound is tighter, and is derived by exploiting the Lipschitz continuity of the discriminator with respect to its model parameters (weight matrices). Similar idea is used in Zhang et al. (2017), however, we derive sharper Lipschitz constants [3] by the key step of decoupling the spectral norms of weight matrices and the number of parameters, i.e., separating $\beta$ and $d^2L$.*

**Remark 4.** *Theorem 2 shows the advantage of spectrum control in generalization by constraining the class of discriminators. However, as suggested in Arora et al. (2017), the class of discriminators needs to be large enough to detect lack of diversity. Despite of a lack of theoretical justifications, empirical results in Miyato et al. (2018) show that discriminators with spectral normalization are powerful in distinguishing $\nu_n$ from $\mu$, and suffer less from the mode collapse. We conjecture that the observed singular value decay (as illustrated in Figure 1) contributes to preventing mode collapse. We leave this for future theoretical investigation.*

## 4 EXPERIMENT

To demonstrate our proposed new methods, we conduct experiments on CIFAR-10 (Krizhevsky & Hinton, 2009), STL-10 (Coates et al., 2011), and ImageNet (Russakovsky et al., 2015). We illustrate the importance of spectrum control in GANs training by revealing a close relation between the performance and the singular value decays.

All implementations are done in Chainer as the official implementation of the SN-GAN (Miyato et al., 2018). Note that SN-GAN is using power iteration. If not specified, all orther Spectral Normalization (SN) methods are under SVD framework. For quantitative assessment of generated examples, we use *inception score* (Salimans et al., 2016) and *Fréchet inception distance* (FID, Heusel et al. (2017)). All reported results correspond to 10 runs of the GAN training with different random initializations. The discussion of this paper is based on fully connected layer. When dealing with convolutional layer, we only need to reshape the 4D weight tensor to a 2D matrix. Denote the weight tensor of a convolutional layer as $W^C \in \mathbb{R}^{c_o, c_i, k_h, k_w}$, where $c_o, c_i, (k_h, k_w)$ denotes the output channel, the input channel and the kernel size. We reshape $W^C$ as $W \in \mathbb{R}^{c_o, c_i \times k_h \times k_w}$ (Huang et al., 2017), i.e., merging the last three dimensions while preserving the first dimension. See more implementation details in Appendix C.1.

---

[3]The Lipschitz constant in Zhang et al. (2017) can be of the order $d^L$.

### 4.1  DC-GAN

We test our methods on DC-GANs with two datasets, CIFAR-10 and STL-10. Specifically, we adopt a 5-layer CNN as the generator and a 7-layer CNN as the discriminator. Recall that our proposed training framework tries to solve the equilibrium for equation (5). We set $\phi(\cdot) = \log(\cdot)$ and $\mathcal{A}$ being the sigmoid function. Denote $f_D(\mathcal{E}, \mathcal{U}, \mathcal{V}) = f(\theta, \mathcal{E}, \mathcal{U}, \mathcal{V}) - \lambda \mathcal{L}_{\text{orth}} - \gamma \mathcal{R}(\mathcal{E})$ for a fixed $\theta$ and $f_G(\theta) = -\mathbb{E}_{x \sim \mathcal{D}_{G_\theta}}[\mathcal{A}(D(x))]$ for fixed $\mathcal{U}, \mathcal{V}$, and $\mathcal{E}$, where $\mathcal{L}_{\text{orth}}(\mathcal{U}, \mathcal{V}) = \sum_{i=1}^{L}(\|U_i^\top U_i - I_i\|_F^2 + \|V_i^\top V_i - I_i\|_F^2)$. We maximize $f_D(\mathcal{E}, \mathcal{U}, \mathcal{V})$ for $n_{\text{dis}}$ iterations ($n_{\text{dis}} \geq 1$) followed by minimizing $f_G(\theta)$ for one iteration. Note that we use a $\log D$ trick (Goodfellow et al., 2014) to ease the computation of minimizing $f_G(\theta)$. Detailed implementations are provided in Appendices B and C.2. We choose tuning parameters $\lambda = 10$ and $\gamma = 1$ in all the experiments except for the Divergence regularizer, where we pick $\lambda = 10$ and $\gamma = 0.05^4$. $\gamma$ is chosen according to the output range of different regularizers. We set a smaller gamma for Divergence Regularizer, since its output is much larger than other regularizers. We take 100K iterations in all the experiments on CIFAR-10 and 200K iterations on STL-10 as suggested in Miyato et al. (2018).

To solve (5), we adopt the setting in Radford et al. (2015), which has been shown to be robust for different GANs by Miyato et al. (2018). Specifically, we use the Adam optimizer (Kingma & Ba, 2014) with the following hyperparameters: (1) $n_{\text{dis}} = 1$; (2) $\alpha = 0.0002$, the initial learning rate; (3) $\beta_1 = 0.5, \beta_2 = 0.999$, the first and second order momentum parameters of Adam respectively.

Before we present our results, we show the effectiveness of our proposed reparameterization, which aims to approximate the singular values of weight matrices while avoiding direct SVDs. As can be seen, in Table 1, $U_i$ and $V_i$ have nearly orthonormal columns respectively, i.e., $\|U_i^\top U_i - I_i\|_F^2, \|V_i^\top V_i - I_i\|_F^2 \leq 10^{-4}$. Although the reparameterization introduces more model parameters, it maintains comparable computational efficiency. See more details in Appendix D.2.

Table 1: The sub-orthogonality of $U_i$'s and $V_i$'s in the discriminator with the divergence regularizer on CIFAR-10 after 100K iterations. For other settings, we also observe that all $U_i$'s and $V_i$'s have nearly orthonormal columns.

|  | Layer 0 | Layer 1 | Layer 2 | Layer 3 | Layer 4 | Layer 5 | Layer 6 |
|---|---|---|---|---|---|---|---|
| $\|U^\top U - I\|_F^2$ | 2.3e-5 | 1.2e-5 | 1.5e-5 | 1.6e-5 | 2.7e-5 | 2.5e-5 | 2.1e-5 |
| $\|V^\top V - I\|_F^2$ | 7.9e-5 | 1.0e-5 | 1.7e-5 | 2.5e-5 | 4.1e-5 | 7.1e-5 | 3.9e-5 |

Figure 4 shows that the singular value decays of weight matrices with two different methods: SN-GAN and $D$-optimal regularizer with spectral normalization. As can be seen, our method achieves a slower decay in singular values than that of SN-GAN. See more results of other methods in Appendix D.3. Such a slower decay improves the performance of GANs. Specifically, Table 2 presents the inception scores and FIDs of our proposed methods as well as other methods on CIFAR-10 and STL-10. As can be seen, under CNN architecture, our methods achieve significant improvements on STL-10. Compared with STL-10, CIFAR-10 is easy to learn, and thus GAN training can only limitedly benefits from encouraging the slow singular value decay. As a result, on CIFAR-10, our methods slightly improve the result of SN-GAN. Moreover, as

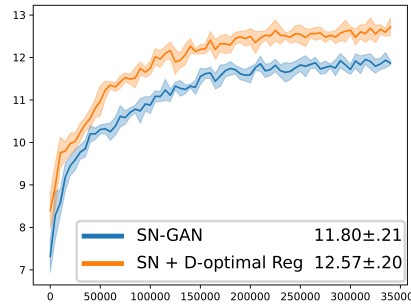

Figure 3: Inception scores on ImageNet. We can see that our method outperforms SN-GAN.

shown in Figure 4, at the early stage (5k iteration), our method achieves slow decay while SN-GAN still decays fast. Thus, it converge faster than SN-GAN as shown in Figure 5.

---

[4]In fact, the performance is not sensitive to these hyperparameters, since we only observe negligible difference by fine tuning these parameters. Specifically, when $\lambda \in [1, 100]$ and $\gamma \in [0.2, 5]$ ($\gamma \in [0.01, 0.1]$ for Divergence regularizer), the algorithm yields similar results.

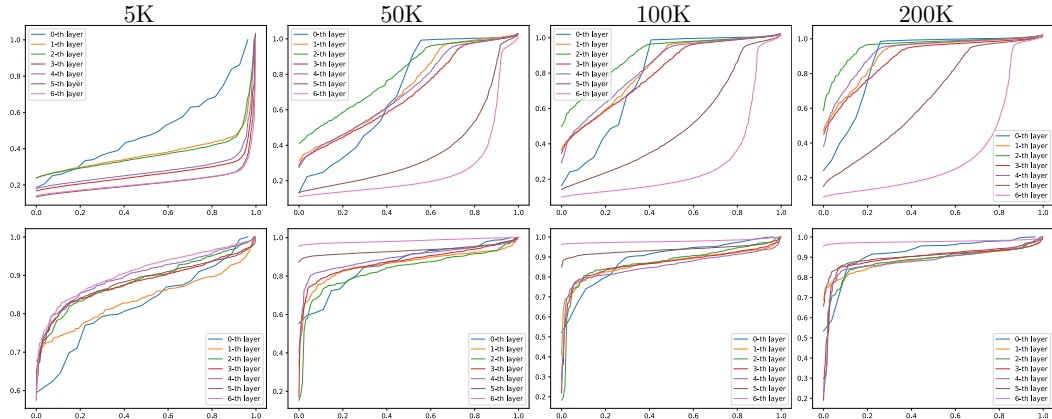

Figure 4: Illustrations of singular value decay in 7 layers at 5K-th, 50K-th, 100K-th, and 200K-th iteration. The above figures are for the SN-GANs; the below are for $D$-optimal regularizer with SN.

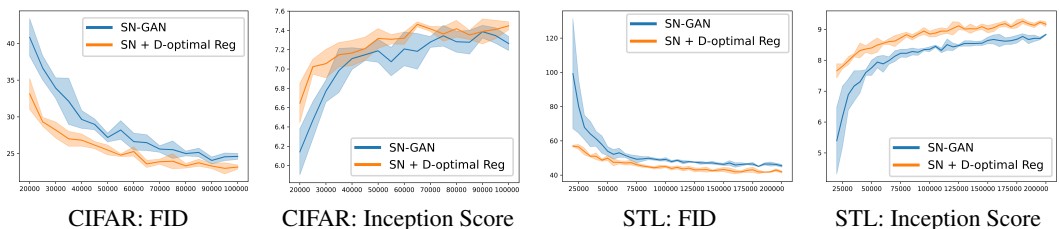

Figure 5: The inception scores and FID's with error bar over 10 runs. Due to the space limit we only present the comparisaon between SN-GAN and D-optimial regularizer with SN, which is the best among our proposed methods. The full comparison with all proposed methods is in Appendix D.4.

## 4.2 RESNET-GAN

We also test our proposed method on ResNet, a more advanced structure, on both discriminator and generator (Appendix C.2). For these experiments, we adopt the hinge loss for adversarial training on discriminators:

$$f_D(\mathcal{E}, \mathcal{U}, \mathcal{V}) = \mathbb{E}_{x \sim \mathcal{D}_{G_\theta}}[\min(0, -1 - D(x))] + \frac{1}{n}\sum_{i=1}^{n} \min(0, -1 + D(x_i)).$$

We also adopt the commonly used hyperparameter settings for the Adam optimizer on ResNet: $n_{\text{dis}} = 5, \alpha = 0.0002, \beta_1 = 0$, and $\beta_2 = 0.9$ (Gulrajani et al., 2017). Due to our computational resource limit, we only test the method of spectral normalization (our version) with $D$-optimal regularizer, which achieves the best performance on CNN experiments. We also test on the official subsampled $64 \times 64$ ImageNet data using the conditional GAN with a projected discriminator Miyato & Koyama (2018).

The results of our experiments on CIFAR-10 and STL-10 are listed in Table 2, and results on ImageNet are shown in Figure 3. We see that our method is much better than the other methods on STL-10 and ImageNet. As for CIFAR-10, our method is better than orthogonal regularizer but slightly worse than SN-GAN. We believe the reason behind is that CIFAR-10 is relatively easy. As can be seen, for CIFAR-10, the inception scores of all methods are around 8, while the inception score of real data is around 11. In contrast, for STL-10, the inception score of real data is around 26, while inception scores of all methods are less than 10. As a result, when the dataset is complicated and network needs high capacity, our method performs better than SN-GAN.

Table 2: The inception scores and FIDs on CIFAR-10 and STL-10. For consistency, we reimplement baselines under our Chainer environment.

| Method | Inception Score | | FID | |
|---|---|---|---|---|
| | CIFAR-10 | STL-10 | CIFAR-10 | STL-10 |
| Real Data | $11.24 \pm .12$ | $26.08 \pm .26$ | 7.8 | 7.9 |
| **CNN Baseline** | | | | |
| WGAN-GP | $6.72 \pm .11$ | $8.42 \pm .09$ | $39.0 \pm .29$ | $54.1 \pm .35$ |
| Orthogonal Reg. | $7.31 \pm .09$ | $8.77 \pm .07$ | $25.7 \pm .33$ | $44.5 \pm .30$ |
| SN-GAN (Power Iter.) | $7.39 \pm .05$ | $8.83 \pm .07$ | $24.7 \pm .25$ | $45.5 \pm .34$ |
| **Ours CNN (Under SVD)** | | | | |
| Spectral Norm. | $7.35 \pm .05$ | $8.69 \pm .08$ | $25.2 \pm .22$ | $44.8 \pm .39$ |
| Spectral Constraint | $7.43 \pm .08$ | $8.97 \pm .05$ | $24.8 \pm .30$ | $44.0 \pm .42$ |
| Lipschitz Reg. | $7.43 \pm .08$ | $8.99 \pm .06$ | $24.1 \pm .28$ | $45.3 \pm .38$ |
| SC + Divergence Reg. | $7.44 \pm .05$ | $9.21 \pm .09$ | $24.3 \pm .21$ | $41.9 \pm .37$ |
| SN + $D$-Optimal Reg. | $7.48 \pm .06$ | $9.25 \pm .08$ | $23.0 \pm .27$ | $40.5 \pm .41$ |
| **ResNet Structure** | | | | |
| Orthogonal Reg. | $7.90 \pm .05$ | $8.83 \pm .05$ | $22.3 \pm .26$ | $44.9 \pm .35$ |
| SN-GAN (Power Iter.) | $8.21 \pm .05$ | $9.15 \pm .06$ | $19.5 \pm .22$ | $43.0 \pm .44$ |
| SN + $D$-Optimal Reg. | $8.06 \pm .06$ | $9.65 \pm .06$ | $20.5 \pm .18$ | $39.9 \pm .33$ |

## 5 CONCLUSION

In this paper, we propose a new SVD-type reparameterization for weight matrices of the discriminator in GANs, allowing us to efficiently manipulate the spectra of weight matrices. We than establish a new generalization bound of GAN to justify the importance of spectrum control on weight matrices. Moreover, we propose new regularizers to encourage the slow singular value decay. Our experiments on CIFAR-10, STL-10, and ImageNet datasets support our proposed methods, theory, and discoveries.

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

# A PROOF IN SECTION 3

## A.1 PROOF OF THEOREM 2

*Proof.* We bound the output of $D(\cdot)$ as follows,

$$\left|D(x)\right| \leq \|W_L\|_2 \|\sigma_{L-1}(\cdots \sigma_1(W_1 x) \cdots)\|_2 \leq \cdots \leq B_x \prod_{i=1}^{L} B_{W_i}.$$

Consider $d_{\mathcal{F},\phi}(\mu, \nu_n) - \inf_{\nu \in \mathcal{D}_G} d_{\mathcal{F},\phi}(\mu, \nu)$. We have

$$d_{\mathcal{F},\phi}(\mu, \nu_n) - \inf_{\nu \in \mathcal{D}_G} d_{\mathcal{F},\phi}(\mu, \nu)$$

$$= d_{\mathcal{F},\phi}(\mu, \nu_n) - d_{\mathcal{F},\phi}(\widehat{\mu}_n, \nu_n) + d_{\mathcal{F},\phi}(\widehat{\mu}_n, \nu_n) - \inf_{\nu \in \mathcal{D}_G} d_{\mathcal{F},\phi}(\widehat{\mu}_n, \nu)$$

$$+ \inf_{\nu \in \mathcal{D}_G} d_{\mathcal{F},\phi}(\widehat{\mu}_n, \nu) - \inf_{\nu \in \mathcal{D}_G} d_{\mathcal{F},\phi}(\mu, \nu)$$

$$\leq 2 \left( \sup_{\mathcal{A}D(\cdot) \in \mathcal{F}} \mathbb{E}_{x \sim \mu}[\phi(\mathcal{A}(D(x)))] - \mathbb{E}_{x \sim \mu_n}[\phi(\mathcal{A}(D(x)))] \right) + \epsilon. \tag{10}$$

Note that given $x_1, \ldots, x_i, \ldots, x_n$ and $x_1, \ldots, x_i', \ldots, x_n$, we have

$$\left| \sup_{\mathcal{A}D(\cdot) \in \mathcal{F}} \mathbb{E}_{x \sim \mu}[\phi(\mathcal{A}(D(x)))] + \mathbb{E}_{x \sim \mu_n}[\phi(\mathcal{A}(D(x)))] \right.$$

$$\left. - \sup_{\mathcal{A}D(\cdot) \in \mathcal{F}} \mathbb{E}_{x \sim \mu}[\phi(\mathcal{A}(D(x)))] + \mathbb{E}_{x \sim \mu_n'}[\phi(\mathcal{A}(D(x)))] \right|$$

$$\leq \frac{\left| \phi(\mathcal{A}(D(x_i))) - \phi(\mathcal{A}(D(x_i'))) \right|}{n}$$

$$\leq \rho_\phi \frac{\left| D(x_i) - D(x_i') \right|}{n}$$

$$\leq \frac{2}{n} \rho_\phi B_x \prod_{i=1}^{L} B_{W_i}.$$

Then McDiarmid's inequality gives us, with probability at least $1 - \delta/2$,

$$\sup_{\mathcal{A}D(\cdot) \in \mathcal{F}} \mathbb{E}_{x \sim \mu}[\phi(\mathcal{A}(D(x)))] - \mathbb{E}_{x \sim \mu_n}[\phi(\mathcal{A}(D(x)))]$$

$$\leq \mathbb{E} \left[ \sup_{\mathcal{A}D(\cdot) \in \mathcal{F}} \mathbb{E}_{x \sim \mu}[\phi(\mathcal{A}(D(x)))] - \mathbb{E}_{x \sim \mu_n}[\phi(\mathcal{A}(D(x)))] \right] + 2\rho_\phi B_x \prod_{i=1}^{L} B_{W_i} \sqrt{\frac{\log \frac{2}{\delta}}{2n}}. \tag{11}$$

By the argument of symmetrization, we have

$$\mathbb{E} \left[ \sup_{\mathcal{A}D(\cdot) \in \mathcal{F}} \mathbb{E}_{x \sim \mu}[\phi(\mathcal{A}(D(x)))] - \mathbb{E}_{x \sim \mu_n}[\phi(\mathcal{A}(D(x)))] \right]$$

$$\leq 2 \mathbb{E}_{x_i \sim \mu, \epsilon} \left[ \frac{1}{n} \sup_{\mathcal{A}D(\cdot) \in \mathcal{F}} \sum_{i=1}^{n} \epsilon_i \phi(\mathcal{A}(D(x_i))) \right], \tag{12}$$

where $\epsilon_i$'s are i.i.d. Rademacher random variables, i.e., $\mathbb{P}(\epsilon_i = 1) = \mathbb{P}(\epsilon_i = -1) = 1/2$. McDiarmid's inequality again gives us, with probability at least $1 - \delta/2$, we have

$$\mathbb{E}_{x_i \sim \mu, \epsilon} \left[ \frac{1}{n} \sup_{\mathcal{A}D(\cdot) \in \mathcal{F}} \sum_{i=1}^{n} \epsilon_i \phi(\mathcal{A}(D(x_i))) \right]$$

$$\leq \mathbb{E}_\epsilon \left[ \frac{1}{n} \sup_{\mathcal{A}D(\cdot) \in \mathcal{F}} \sum_{i=1}^{n} \epsilon_i \phi(\mathcal{A}(D(x_i))) \right] + 2\rho_\phi B_x \prod_{i=1}^{L} B_{W_i} \sqrt{\frac{\log \frac{2}{\delta}}{2n}}. \tag{13}$$

Note that $\mathbb{E}_\epsilon \left[ \frac{1}{n} \sup_{\mathcal{A}D(\cdot) \in \mathcal{F}} \sum_{i=1}^{n} \epsilon_i \phi(\mathcal{A}(D(x))) \right]$ is essentially the empirical Rademacher complexity of $\phi(\mathcal{A}(D(\cdot)))$. Since $\phi$ and $\mathcal{A}$ are both Lipschitz, by Talagrand's lemma, we have

$$\mathbb{E}_\epsilon \left[ \frac{1}{n} \sup_{\mathcal{A}D(\cdot) \in \mathcal{F}} \sum_{i=1}^{n} \epsilon_i \phi(\mathcal{A}(D(x_i))) \right] \leq \rho_\phi \mathbb{E}_\epsilon \left[ \frac{1}{n} \sup_{D} \sum_{i=1}^{n} \epsilon_i D(x_i) \right].$$

We then use the standard Dudley's entropy integral to bound $\mathbb{E}_\epsilon \left[ \frac{1}{n} \sup_D \sum_{i=1}^n \epsilon_i D(x_i) \right]$. We exploit the parametric form of discriminators to find a tight covering number. We have to investigate the Lipschitz continuity of $D(\cdot)$ with respect to the weight matrices $W_1, \ldots, W_L$. We based our argument on telescoping. Given two sets of weight matrices $W_1, \ldots, W_L$ and $W_1', \ldots, W_L'$ and fix the activation operators and $\mathcal{A}$, we have

$$\|D(x) - D'(x)\|_\infty$$
$$\leq \|W_L \sigma_{L-1}(\cdots \sigma_1(W_1 x) \cdots) - W_L' \sigma_{L-1}(\cdots \sigma_1(W_1' x) \cdots)\|_2$$
$$= \|W_L \sigma_{L-1}(\cdots \sigma_1(W_1 x) \cdots) - W_L' \sigma_{L-1}(\cdots \sigma_1(W_1 x) \cdots)\|_2$$
$$\quad + \|W_L' \sigma_{L-1}(\cdots \sigma_1(W_1 x) \cdots) W_L' \sigma_{L-1}(\cdots \sigma_1(W_1' x) \cdots)\|_2$$
$$\leq \|W_L - W_L'\|_2 \|\sigma_{L-1}(\cdots \sigma_1(W_1 x) \cdots)\|_2$$
$$\quad + \|W_L'\|_2 \|\sigma_{L-1}(\cdots \sigma_1(W_1 x) \cdots) - \sigma_{L-1}(\cdots \sigma_1(W_1' x) \cdots)\|_2$$
$$\leq \|W_L - W_L'\|_2 B_x \prod_{i=1}^{L-1} B_{W_i} + \|W_L'\|_2 \|\sigma_{L-1}(\cdots \sigma_1(W_1 x) \cdots) - \sigma_{L-1}(\cdots \sigma_1(W_1' x) \cdots)\|_2$$
$$\leq \cdots \cdots$$
$$\leq \sum_{i=1}^L \frac{B_x \prod_{j=1}^L B_{W_j}}{B_{W_i}} \|W_i - W_i'\|_2.$$

For notational simplicity, we denote $L_{W_i} = \frac{B \prod_{j=1}^L B_{W_i}}{B_{W_i}}$. When the activation operators and $\mathcal{A}$ are given, function $D$ has a one to one correspondence to weight matrices $W_1, \ldots, W_L$. Thus, to construct a covering of $\mathcal{F}$, it is enough to construct matrix coverings of $W_1, \ldots, W_L$, and their Cartesian product gives us a covering of $\mathcal{F}$. The standard argument of volume ratio gives us an upper bound of the covering number of matrices with bounded spectral norms. Suppose $\mathcal{M} = \{ M \in \mathbb{R}^{d \times h} : \|M\|_2 \leq \lambda \}$, the covering number $\mathcal{N}(\mathcal{M}, \epsilon, \|\cdot\|_2)$ at scale $\epsilon$ with respect to spectral norm is bounded by

$$\mathcal{N}(\mathcal{M}, \epsilon, \|\cdot\|_2) \leq \left( 1 + \frac{\min(\sqrt{d}, \sqrt{h}) \lambda}{\epsilon} \right)^{dh}.$$

Therefore, the covering number $\mathcal{N}(\mathcal{F}, \epsilon, \|\cdot\|_\infty)$ is bounded by

$$\mathcal{N}(\mathcal{F}, \epsilon, \|\cdot\|_\infty) \leq \prod_{i=1}^L \mathcal{N}(W_i, \frac{\epsilon}{L L_{W-i}}, \|\cdot\|_2)$$
$$\leq \prod_{i=1}^L \left( 1 + \frac{L L_{W_i} \min(\sqrt{d_i}, \sqrt{d_{i+1}}) B_{W_i}}{\epsilon} \right)^{d_i d_{i+1}}.$$

Take $d = \max\{d_1, \ldots, d_L\}$. We get

$$\mathcal{N}(\mathcal{F}, \epsilon, \|\cdot\|_\infty) \leq \left( 1 + \frac{\sqrt{d} L B_x \prod_{i=1}^L B_{W_i}}{\epsilon} \right)^{d^2 L}.$$

Then Dudley's entropy integral gives us

$$\mathbb{E}_\epsilon \left[ \frac{1}{n} \sup_D \sum_{i=1}^n \epsilon_i D(x_i) \right] \leq \frac{4\alpha}{\sqrt{n}} + \frac{12}{n} \int_\alpha^{B_x \prod_{i=1}^L B_{W_i} \sqrt{n}} \sqrt{\log \mathcal{N}(\mathcal{F}, \epsilon, \|\cdot\|_\infty)} d\epsilon$$

$$\leq \frac{4\alpha}{\sqrt{n}} + \frac{12}{n} B_x \prod_{i=1}^L B_{W_i} \sqrt{n} \sqrt{d^2 L \log \left( 1 + \frac{\sqrt{d} L B_x \prod_{i=1}^L B_{W_i}}{\alpha} \right)}.$$

It is enough to pick $\alpha = \frac{1}{\sqrt{n}}$, which yields

$$\mathbb{E}_\epsilon \left[ \frac{1}{n} \sup_D \sum_{i=1}^n \epsilon_i D(x_i) \right] \leq \frac{4}{n} + \frac{12 B_x \prod_{i=1}^L B_{W_i} \sqrt{d^2 L \log \left( 2\sqrt{dn} L B_x \prod_{i=1}^L B_{W_i} \right)}}{\sqrt{n}}.$$

Thus, we immediately have,

$$\mathbb{E}_\epsilon \left[ \frac{1}{n} \sup_{\mathcal{A}D(\cdot) \in \mathcal{F}} \sum_{i=1}^n \epsilon_i \phi(\mathcal{A}(D(x_i))) \right]$$

$$\leq \frac{4\rho_\phi}{n} + \frac{12\rho_\phi B_x \prod_{i=1}^L B_{W_i} \sqrt{d^2 L \log \left( 2\sqrt{dn} L B_x \prod_{i=1}^L B_{W_i} \right)}}{\sqrt{n}}. \tag{14}$$

Now, combining equations (10), (11), (12), (13), and (14) together, we get

$$d_{\mathcal{F},\phi}(\mu, \nu_n) \leq \inf_{\nu \in \mathcal{D}_G} d_{\mathcal{F},\phi}(\mu, \nu_n) + \frac{16\rho_\phi}{n} + \frac{48\rho_\phi \beta \sqrt{d^2 L \log \left( 2\sqrt{dn} L \beta \right)}}{\sqrt{n}} + 12\rho_\phi \beta \sqrt{\frac{\log \frac{1}{\delta}}{n}},$$

where $\beta = B_x \prod_{i=1}^L B_{W_i}$. On the other hand, naively applying the argument from Bartlett et al. (2017) yields the generalization bound

$$d_{\mathcal{F},\phi}(\mu, \nu_n) \leq \inf_{\nu \in \mathcal{D}_G} d_{\mathcal{F},\phi}(\mu, \nu_n) + O \left( \frac{\rho_\phi \beta \sqrt{d L^3 \log \left( \sqrt{dn} L \beta \right)}}{\sqrt{n}} + \rho_\phi \beta \sqrt{\frac{\log \frac{1}{\delta}}{n}} \right).$$

Combining the two generalization bound together, we get

$$d_{\mathcal{F},\phi}(\mu, \nu_n)$$

$$\leq \inf_{\nu \in \mathcal{D}_G} d_{\mathcal{F},\phi}(\mu, \nu_n) + O \left( \frac{\rho_\phi \beta \sqrt{d L \min(d, L^2) \log \left( \sqrt{dn} L \beta \right)}}{\sqrt{n}} + \rho_\phi \beta \sqrt{\frac{\log \frac{1}{\delta}}{n}} \right). \tag{15}$$

$\square$

# B  ALGORITHM

Recall that we are maximizing the following objective function $f_D(\mathcal{E}, \mathcal{U}, \mathcal{V})$ for discriminator $D$ in Section 4.1:

$$f_D(\mathcal{E}, \mathcal{U}, \mathcal{V}) = f(\theta, \mathcal{E}, \mathcal{U}, \mathcal{V}) - \lambda \mathcal{L}_{orth}(\mathcal{U}, \mathcal{V}) - \gamma \mathcal{R}(\mathcal{E}).$$

The detailed training algorithm is described in Algorithm 1:

---

**Algorithm 1** Adversarial training with Spectrum Control of Discriminator, $D$

---

***Initialization***
 1: **for** $l = 1, .., L$ **do**
 2:     Determine the rank of $l$-layer: $r_i = \min\{d_i, d_{i+1}\}$.
 3:     Initialize $U_i \in \mathbb{R}^{d_i \times r_i}$ and $V_i \in \mathbb{R}^{d_{i+1} \times r_i}$ with orthonormal columns.
 4:     Initialize $E_i = I_{r_i}$.
 5: **end for**

---

***Forward pass***
**Input:** mini-batch input $H_i \in \mathbb{R}^{m \times d_i}$ from previous layer
**Output:** mini-batch output $S_{i+1} \in \mathbb{R}^{m \times d_{i+1}}$
**Parameters:** $U_i, V_i$, and $E_i$
 1: Perform *Singluar value update* on $E_i$
 2: Calculate weight matrix: $W_i = U_i E_i V_i^\top \in \mathbb{R}^{d_i \times d_{i+1}}$.
 3: Calculate output: $S_{i+1} = H_i W_i$.

---

***Backward pass***
**Input:** activation derivative $\nabla_{S^{i+1}} f$
**Output:** $\nabla_{H_i} f, \nabla_{U_i} f_D, \nabla_{V_i} f_D, \nabla_{E_i} f_D$
 1: Calculate: $\nabla_{H_i} f = \nabla_{S_{i+1}} f W_i^\top$ as standard linear module.
 2: Calculate: $\nabla_{W_i} f = H_i^\top \nabla_{S_{i+1}} f$ as standard linear module.
 3: Calculate: $\nabla_{U_i} f, \nabla_{V_i} f, \nabla_{E_i} f$ based on $\nabla_{W_i} f$.
 4: Calculate: $\nabla_{U_i} f_D = \nabla_{U_i} f - \lambda \nabla_{U_i} \mathcal{L}_{\text{orth}}, \nabla_{V_i} f_D = \nabla_{V_i} f - \lambda \nabla_{V_i} \mathcal{L}_{\text{orth}}$.
 5: Calculate: $\nabla_{E_i} f_D = \nabla_{E_i} f - \gamma \nabla_{E_i} \mathcal{R}(\mathcal{E})$.
 6: Update $U_i, V_i$, and $E_i$ with the Adam Optimizer.

---

***Singluar value update***
**Input:** $E_i$
**Output:** $E_i$ with $e_1^i \in [0, 1]$, i.e. the largest singular value is bounded by 1
 1: If use spectrum constraint: $E_i = g(E_i)$, where $g(\cdot)$ is the clipping operator defined in (8).
 2: If use spectrum normalization: $E_i = E_i / e_1^i$.
 3: If use Lipschitz regularizer: do nothing, since we do not need to layer-wisely control the Lipschitz constant in this case.

---

Note that we omit the bias term for simplicity.

## C  EXPERIMENT SETTING

### C.1  PERFORMANCE MEASURE

Inception score is introduced by Salimans et al. (2016):

$$I(\{x_i\}_{i=1}^n) := exp(E[D_{KL}[p(y|x)||p(y)]]),$$

where $p(y)$ is estimated by $\frac{1}{n}\sum_{i=1}^n p(y|x_i)$ and $p(y|x)$ is estimated by a pretrained Inception Net, $f_{incept}$ Szegedy et al. (2015). Following the procedure in Salimans et al. (2016), we calculated the score for randomly generated 5000 examples from generator for 10 times. The average and the standard deviation of the inception scores are reported.

Fréchet inception distance (FID) is introduced by Heusel et al. (2017). FID uses 2nd order information of the final layer of the inception model applied to the examples. To begin with, *Fréchet distance* (FD, Dowson & Landau (1982)) is 2-Wasserstein distance between two Gaussian distribution $p_1$ and $p_2$:

$$F(p_1, p_2) = \|\boldsymbol{\mu}_1 - \boldsymbol{\mu}_2\|_2^2 + tr[\Sigma_1 + \Sigma_2 - 2(\Sigma_1\Sigma_2)^{1/2}],$$

where $\{\boldsymbol{\mu}_1, \Sigma_1\}$ and $\{\boldsymbol{\mu}_2, \Sigma_2\}$ are the mean and covariance of $p_1$ and $p_2$ respectively. FID between two image distribution $p_1$ and $p_2$ is the FD between $f_{incept}(p_1)$ and $f_{incept}(p_2)$, i.e., the distribution after the inception net transformation. The emperical FID is calculated by sampling 10000 true images and 5000 images from generator, $\mathcal{D}_{G_\theta}$. Different from inception score, multiple repetition of the experiments did not exhibit any notable variations on this score.

Acknowledging that different realizations of Inception Net results in different inception scores (Barratt & Sharma, 2018), we test inception scores with the standard tensorflow inception net for consistency.

## C.2 NETWORK ARCHITECTURE

Table 3: The standard CNN architecture for CIFAR-10 and STL-10. For CIFAR-10, $M = 32$, $M_g = 4$. While for STL-10, $M = 48$, $M_g = 6$. The slopes coefficient is 0.1 for all LeakyReLU activations.

(a) Generator

| | |
|---|---|
| **Input**: $z \in \mathbb{R}^{128} \sim \mathcal{N}(0, I)$ | |
| Linear: $128 \to M_g \times M_g \times 512$ | |
| Deconv: $[4 \times 4, 256, \text{stride} = 2]$ | BN, ReLU |
| Deconv: $[4 \times 4, 128, \text{stride} = 2]$ | BN, ReLU |
| Deconv: $[4 \times 4, 64, \text{stride} = 2]$ | BN, ReLU |
| Conv: $[3 \times 3, 3, \text{stride} = 1]$ | Tanh |

(b) Discriminator

| | |
|---|---|
| **Input**: Image $x \in \mathbb{R}^{M \times M \times 3}$ | |
| Conv: $[3 \times 3, 64, \text{stride} = 1]$ | LeakyReLU |
| Conv: $[4 \times 4, 64, \text{stride} = 2]$ | LeakyReLU |
| Conv: $[3 \times 3, 128, \text{stride} = 1]$ | LeakyReLU |
| Conv: $[4 \times 4, 128, \text{stride} = 2]$ | LeakyReLU |
| Conv: $[3 \times 3, 256, \text{stride} = 1]$ | LeakyReLU |
| Conv: $[4 \times 4, 256, \text{stride} = 2]$ | LeakyReLU |
| Conv: $[3 \times 3, 512, \text{stride} = 1]$ | LeakyReLU |
| Linear: $M_g \times M_g \times 512 \to 1$ | |

Table 4: The ResNet architectures for CIFAR-10 and STL-10 datasets.

(a) CIFAR-10 Generator

| | |
|---|---|
| **Input**: | $z \in \mathbb{R}^{128} \sim \mathcal{N}(0, I)$ |
| Linear: | $128 \to 4 \times 4 \times 256$ |
| ResBlocks: | [256, Up-sampling] $\times 3$ |
| | BN,ReLU |
| Conv: | $[3 \times 3, 3, \text{stride} = 1]$, Tanh |

(b) CIFAR-10 Discriminator

| | |
|---|---|
| **Input**: | Image $x \in \mathbb{R}^{32 \times 32 \times 3}$ |
| ResBlocks: | [128, Down-Sampling] $\times 2$ |
| ResBlocks: | [128] $\times 2$ |
| | ReLU, Global sum pooling |
| Linear: | $4 \times 4 \times 128 \to 1$ |

(c) STL-10 Generator

| | |
|---|---|
| **Input**: | $z \in \mathbb{R}^{128} \sim \mathcal{N}(0, I)$ |
| Linear: | $128 \to 6 \times 6 \times 512$ |
| ResBlock: | [256, Up-sampling] |
| ResBlock: | [128, Up-sampling] |
| ResBlock: | [64, Up-sampling] |
| | BN,ReLU |
| Conv: | $[3 \times 3, 3, \text{stride} = 1]$, Tanh |

(d) STL-10 Discriminator

| | |
|---|---|
| **Input**: | Image $x \in \mathbb{R}^{48 \times 48 \times 3}$ |
| ResBlock: | [64, Down-Sampling] |
| ResBlock: | [128, Down-Sampling] |
| ResBlock: | [256, Down-Sampling] |
| ResBlock: | [512, Down-Sampling] |
| ResBlock: | [1024] |
| | ReLU, Global sum pooling |
| Linear: | $3 \times 3 \times 128 \to 1$ |

Table 5: The ResNet architectures for ImageNet dataset. Recall that we adopts conditional GAN framework with projection discriminator. The ResBlock is implemented with the conditional batch normalization for the generator. $\langle Embed(y), h \rangle$ is the inner product of label embedding, $Embed(y)$, and the hidden state, $h$, after the global sum pooling. (Miyato & Koyama, 2018). We use the same Residual Block as Gulrajani et al. (2017) describes.

(a) Generator

| **Input**: | $z \in \mathbb{R}^{128} \sim \mathcal{N}(0, I)$ |
|---|---|
| Linear: | $128 \rightarrow 4 \times 4 \times 1024$ |
| ResBlock: | [1024, Up-sampling] |
| ResBlock: | [512, Up-sampling] |
| ResBlock: | [128, Up-sampling] |
| ResBlock: | [64, Up-sampling] |
| | BN,ReLU |
| Conv: | [$3 \times 3$, 3, stride = 1], Tanh |

(b) Discriminator

| **Input**: | Image $x \in \mathbb{R}^{64 \times 64 \times 3}$ |
|---|---|
| | Label $y \in \{1, 2, 3, ..., 1000\}$ |
| ResBlock: | [64, Down-Sampling] |
| ResBlock: | [128, Down-Sampling] |
| ResBlock: | [256, Down-Sampling] |
| ResBlock: | [512, Down-Sampling] |
| ResBlock: | [1024, Down-Sampling] |
| | ReLU, Global sum pooling |
| Projection+ Linear | : $\langle Embed(y), \mathsf{h} \rangle + [1024 \rightarrow 1]$ |

# D  SUPPLEMENTARY RESULTS

## D.1  ACCURACY OF APPROXIMATING SINGULAR VALUE DECOMPOSITION

In this section we evaluate how accurate can $E_i = \{e_j^i\}_{j=1}^{r_i}$ approximate the true singular value $\widetilde{E}_i = \{\widetilde{e}_j^i\}_{j=1}^{r_i}$ of recovered weight matrix $W_i = U_i E_i V_i^{\top}$. In Table 6, we compare the maximum, the minimum, the mean and the variance of $E_i$ and $\widetilde{E}_i$.

Table 6: The accuracy of singular value estimation on CIFAR-10 experiment with divergence regularizer after 100K iterations. For other layers and other setting, we can also observe highly accurate singular value approximation.

| | Max | | Min | | Mean | | Var | | $\|U^{\top}U - I\|_F^2$ | $\|V^{\top}V - I\|_F^2$ |
|---|---|---|---|---|---|---|---|---|---|---|
| | $E_i$ | $\widetilde{E}_i$ | $E_i$ | $\widetilde{E}_i$ | $E_i$ | $\widetilde{E}_i$ | $E_i$ | $\widetilde{E}_i$ | | |
| 0-th Layer | 1.0000 | 1.0000 | 0.8344 | 0.8344 | 0.9448 | 0.9448 | 0.0037 | 0.0037 | 2.3e-5 | 7.9e-5 |
| 1-th Layer | 1.0000 | 0.9999 | 0.8128 | 0.8128 | 0.9441 | 0.9441 | 0.0036 | 0.0036 | 1.2e-5 | 1.0e-5 |
| 2-th Layer | 1.0000 | 0.9999 | 0.7972 | 0.7971 | 0.9438 | 0.9438 | 0.0036 | 0.0036 | 1.5e-5 | 1.7e-5 |
| 3-th Layer | 1.0000 | 1.0000 | 0.7969 | 0.7969 | 0.9438 | 0.9438 | 0.0036 | 0.0036 | 1.6e-5 | 2.5e-5 |
| 4-th Layer | 1.0000 | 1.0000 | 0.7816 | 0.7815 | 0.9432 | 0.9432 | 0.0036 | 0.0036 | 2.7e-5 | 4.1e-5 |
| 5-th Layer | 1.0000 | 1.0002 | 0.7817 | 0.7816 | 0.9434 | 0.9434 | 0.0037 | 0.0037 | 2.5e-5 | 7.1e-5 |
| 6-th Layer | 1.0000 | 1.0001 | 0.7765 | 0.7765 | 0.9591 | 0.9591 | 0.0033 | 0.0033 | 2.1e-5 | 3.9e-5 |

## D.2  TIMING COMPARISON

We compare the computational time on CIFAR-10 for different methods in Figure 6. We can see Spectral Constraint with Divergence Regularizer is slightly slower than other methods. Note that the training time will be impacted by the hardware condition, i.e. how many tasks are simultaneously running on the same server. In order to leverage such fluctuation, we test timing for every 1000 iterations and then take their average.

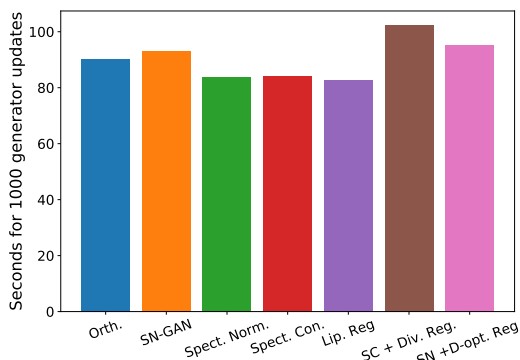

Figure 6: Time cost per 1000 iterations.

## D.3  SINGULAR VALUE DECAY PATTERN

Here, we present an illustration of the singular value decays of weight matrices in 7 layers after 5K, 50K, 100K, and 200K iterations on STL-10 Data for different methods in Table 7. On CIFAR-10, we also observe a very similar pattern, and thus we only present the STL-10 results.

We summarize our observation as follows:
(1) Spectral normalization in SN-GAN yields a slow decay pattern because the power iteration consistently underestimate the spectral norm. So it rescales lower-ranking eigenvalues to br 1. (2) Under SVD reparametrization, the spectral norm is accurate. And thus the eigenvalues decay faster

than the one with power iteration. (3) Because of D-Optimal regularization, the eigenvalues are pushed towards 1 and thus they decay slower. (4) The divergence regularizer pushes the eigenvalues towards a predefined slow decay curve and thus yields such a pattern. (5) The spectral constraint only clip the singluar values which are out of the range $[0, 1]$. It will not rescale the other singular values as the spectral normalization. The singular values tend to grow through the training process. Thus it yields a spectrum with almost no decay. (6) Similar to the spectral constraint, Lipschitz regularizer only penalize the largest singular value of each layer. Thus it yields a spectrum with almost no decay. But it does not enforces the largest singular value of each layer to be 1.

Table 7: Singular value decays.

## D.4 IMAGE GENERATION

For CIFAR-10, the growth of inception score and FID over iterations is shown in Figure 7. While the image generation results are shown in Figure 9.

For STL-10, the growth of inception score and FID over iterations is shown in Figure 8. While the image generation results are shown in Figure 10. For ImageNet, the results are shown in Figure 11

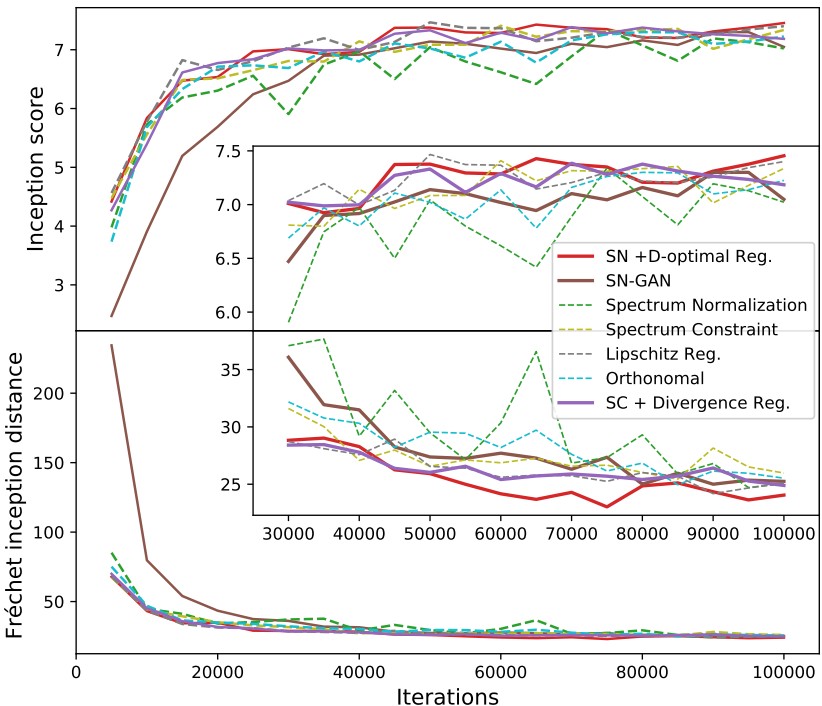

Figure 7: The growth of inception score and FID over iterations on CIFAR-10 dataset.

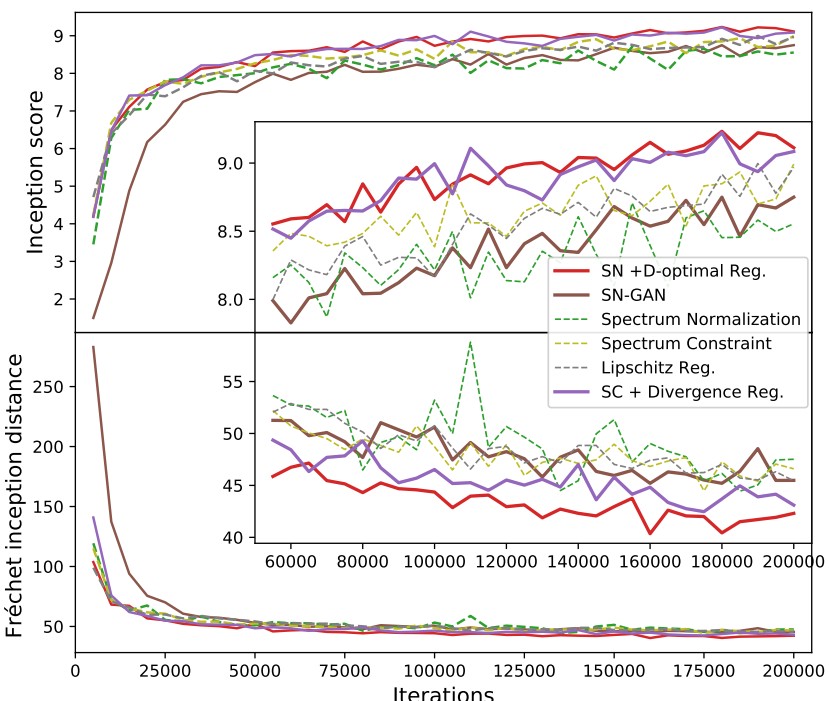

Figure 8: The inception scores and FID's of different methods on STL-10. The top part is for the inception score, and the bottom is for the FID. The inner graph zooms in the results after 55K iterations. The $x$-axis denotes the number of iterations.

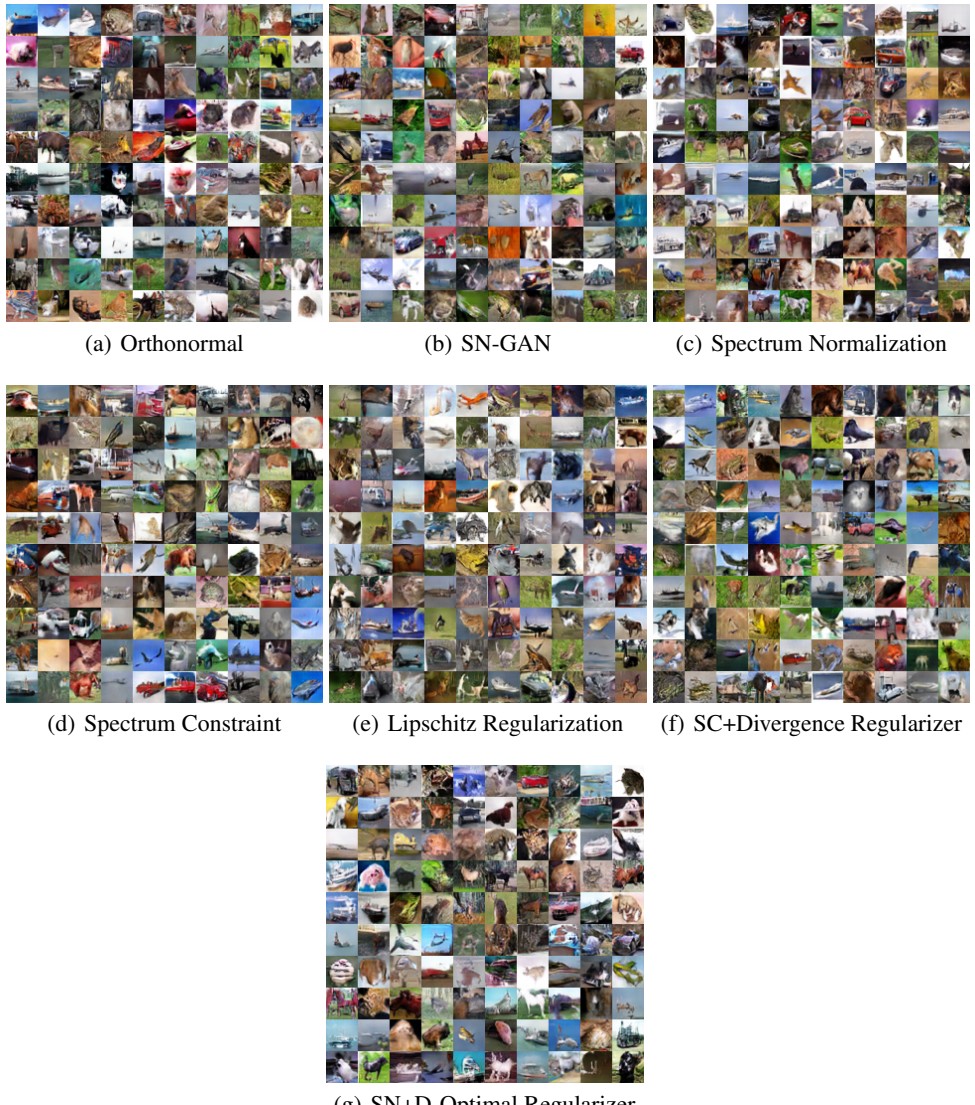

(a) Orthonormal           (b) SN-GAN           (c) Spectrum Normalization

(d) Spectrum Constraint     (e) Lipschitz Regularization     (f) SC+Divergence Regularizer

(g) SN+D-Optimal Regularizer

Figure 9: Image generation on CIFAR-10 dataset.

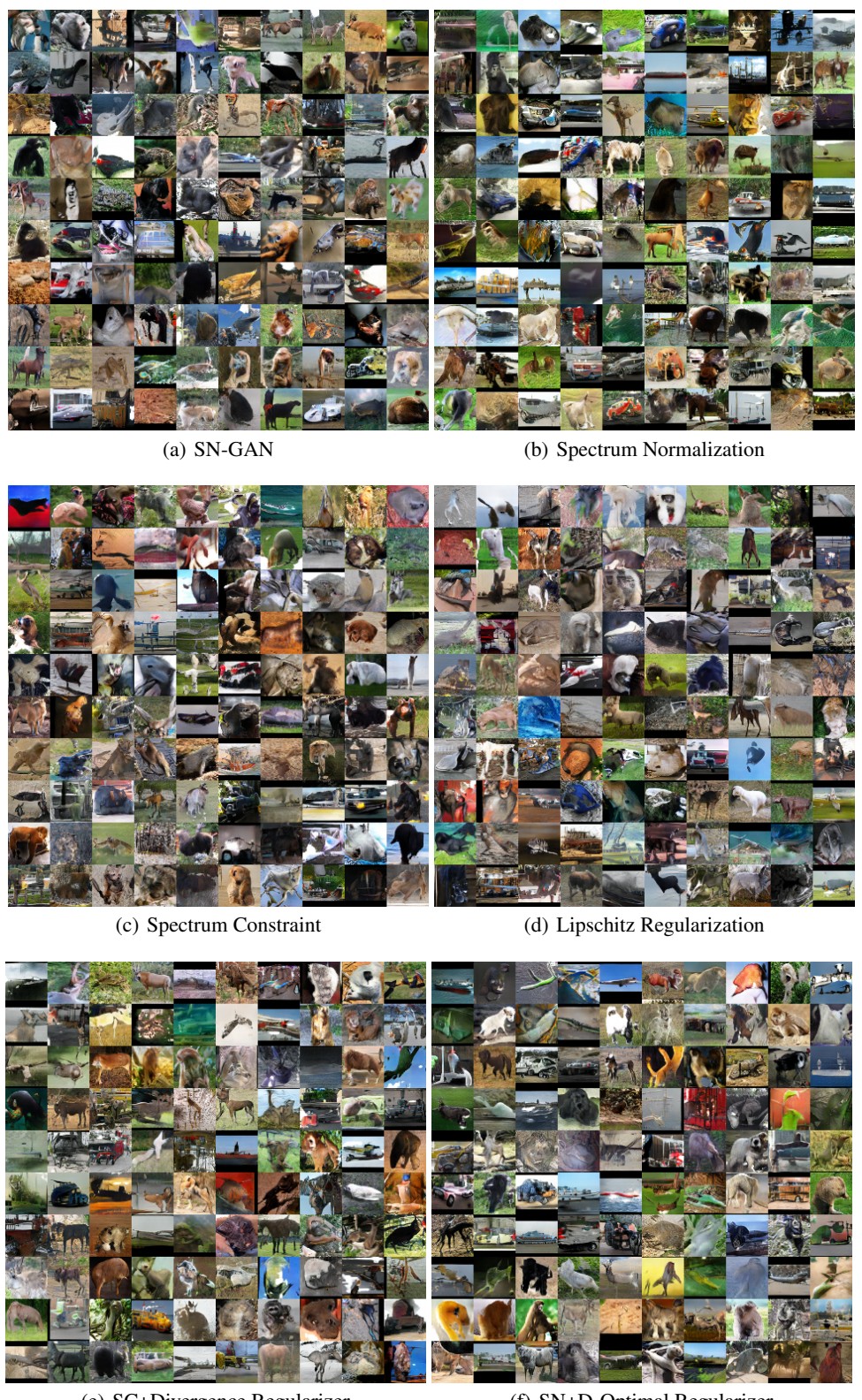

(a) SN-GAN  (b) Spectrum Normalization

(c) Spectrum Constraint  (d) Lipschitz Regularization

(e) SC+Divergence Regularizer  (f) SN+D-Optimal Regularizer

Figure 10: Image generation on STL-10 dataset.

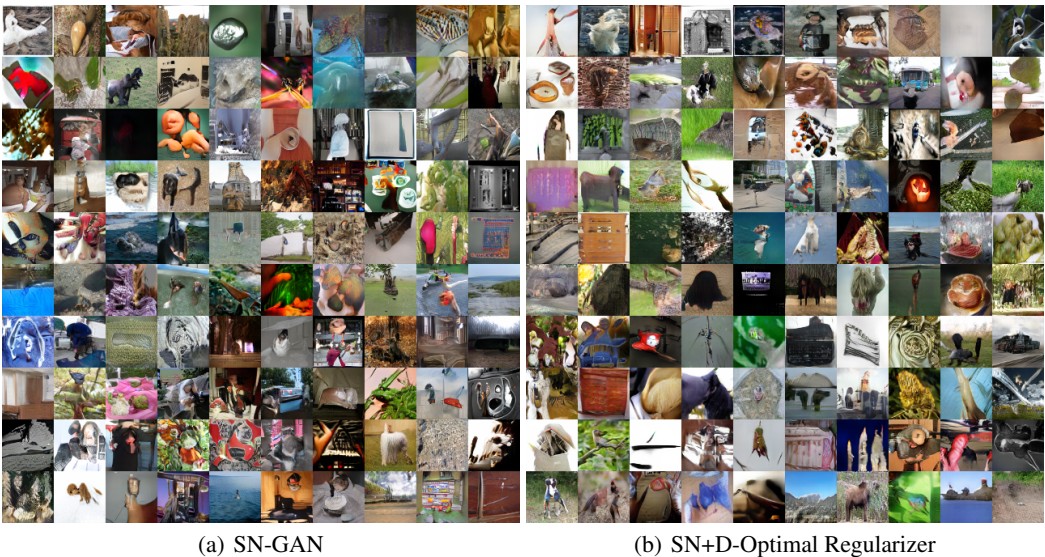

|                      |                              |
| :------------------: | :--------------------------: |
| (a) SN-GAN           | (b) SN+D-Optimal Regularizer |

Figure 11: Image generation on ImageNet.

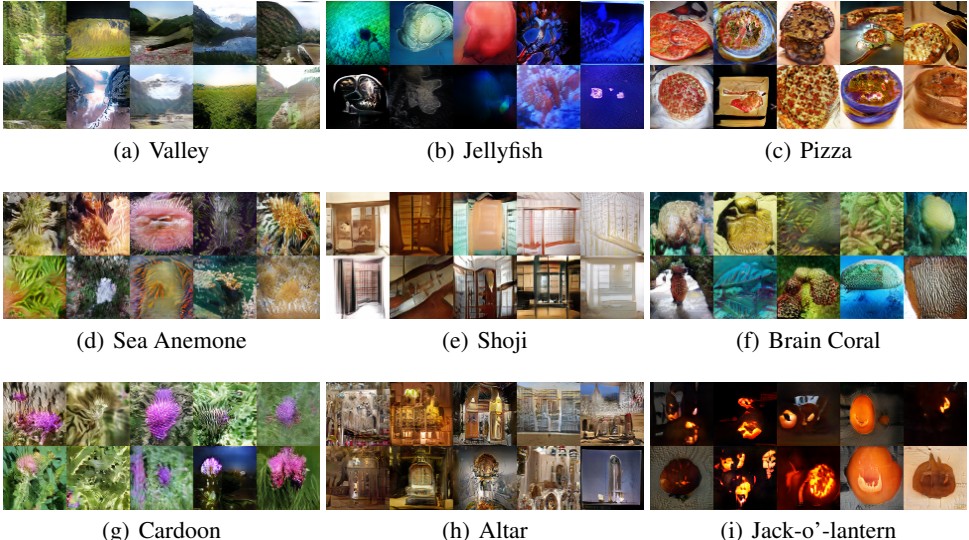

|                     |               |                      |
| :-----------------: | :-----------: | :------------------: |
| (a) Valley          | (b) Jellyfish | (c) Pizza            |
| (d) Sea Anemone     | (e) Shoji     | (f) Brain Coral      |
| (g) Cardoon         | (h) Altar     | (i) Jack-o'-lantern  |

Figure 12: Conditional Image generation on ImageNet (SN+D-Optimal Regularizer)

