# OpenReview forum: "On Computation and Generalization of Generative Adversarial Networks under Spectrum Control"
_ICLR.cc/2019/Conference_

### Official Review · AnonReviewer2 · 2018-10-28
**Reviewer 2 Review**

**Rating:** 7
**Confidence:** 4

**Review:**

This paper proposes to parameterize the weight matrices of neural nets using the SVD, with approximate orthogonality enforced on the singular vectors using Orthogonal Regularization (as opposed to e.g. the Cayley transform or optimizing on the Stiefel manifold), allowing for direct, efficient control over the spectra. The method is applied to GAN discriminators to stabilize training as a natural extension of Spectral Normalization. This method incurs a slight memory and compute cost and achieves a minor performance improvement over Spectral Normalization on two benchmark image generation tasks.

I'm a bit back and forth on this paper. On the one hand, I think the ideas this paper proposes are very interesting and could provide a strong basis off which future work can be built--the extension of spectral normalization to further study and manipulation of the spectra is natural and very promising. However, the results obtained are not particularly strong, and as they stand do not, in my opinion, justify the increased compute and memory cost of the proposed methods. The paper's presentation also wavers between being strong (there were some sections I read and immediately understood) and impenetrable (there were other sections which I had to read 5-10 times just to try and grip what was going on).

Ultimately, my vote is for acceptance. I think that we should not throw out a work with interesting and potentially useful ideas just because it does not set a new SOTA, especially when the current trend with GANs seems to suggest that top performance comes at a compute cost that all but a few groups do not have access to. With another editing pass to improve language and presentation this would be a strong, relevant paper worthy of the attention of the ICLR community.

My notes:

-The key idea of parameterizing matrices as the SVD by construction, but using a regularizer to properly constrain U and V (instead of the expensive Cayley transform, or trying to pin the matrices to the Stifel manifold) is very intriguing, and I think there is a lot of potential here.

-This paper suffers from a high degree of mathiness, substituting dense notation in places where verbal explanation would be more appropriate. There are several spots where explaining the intuition behind a given idea (particularly when proposing the various spectrum regularizers) would be far more effective than the huge amount of notation. In the author's defense, the notation is generally used as effectively as it could be. My issue is that it often is just insufficient, and communication would be better served with more illustrative figures and/or language.

-I found the way the paper references Figure 1 confusing. The decays are substantially different for each layer--are these *all* supposed to be examples of slow decay? Layer 6 appears to have 90% of its singular values below 0.5, while layer 0 has more than 50%. If this is slow decay, what does an undesirable fast decay look like? Isn't the fast decay as shown in figure 2 almost exactly what we see for Layer 6 in figure 1? What is the significance of the sharp drop that occurs after some set number of singular values? The figure itself is easy to understand, but the way the authors repeatedly refer to it as an example of smooth singular decays is confusing.

-what is D-optimal design? This is not something commonly known in the ML literature. The authors should explain what exactly that D-optimal regularizer does, and elucidate its backward dynamics (in an appendix if space does not permit it in the main body). Does it encourage all singular values to have similar values? Does it push them all towards 1? I found the brief explanation ("encourages a slow singular value decay") to be too brief--consider adding  a plot of the D-optimal spectrum to Figure 1, so that the reader can easily see how it would compare to the observed spectra. Ideally, the authors would show an example of the target spectra for each of the proposed regularizers in Figure 1. This might also help elucidate what the authors consider a desirable singular value decay, and mollify some of the issues I take with the way the paper references figure 1.

-The explanation of the Divergence Regularizer is similarly confusing and suffers from mathiness, a fact which I believe is further exacerbated by its somewhat odd motivation. Why, if the end result is a reference curve toward which the spectra will be regularized, do the authors propose (1) a random variable which is a transformation of a gaussian (2) to take the PDF of that random variable (3) discretize the PDF  (4) take the KL between a uniform discrete distribution and the discretized PMF and (5) ignore the normalization term? If the authors were actually working with random variables and proposing a divergence this might make sense, but the items under consideration are singular values which are non-stochastic parameters of a model, so treating them this way seems very odd. Based on figure 2 it looks like the resulting reference curves are fine, but the explanation of how to arrive there is quite convoluted--I would honestly have been more satisfied if the authors had simply designed a function (a polynomial logarithmic function perhaps) with a hyperparameter or two to control the curvature.

-"Our experimental results show that both combinations achieve an impressive results on CIFAR10 and STL-10 datasets"
Please do not use subjective adjectives like "impressive." A 6.5% improvement is okay, but not very impressive, and when you use subjective language you run the risk of readers and reviewers subjectively disagreeing with you, as is the case with this reviewer. Please also fix the typo in this sentence, it should at least be "...achieve [impressive] results" or "achieve an [impressive] improvement on..."

Section 3:
-What is generalization supposed to mean in this context? It's unclear to me why this is at all relevant--is this supposed to be indicating the bounds for which the Discriminator will correctly distinguish real vs generated images? Or is there some other definition of generalization which is relevant? Does it actually matter for what we care about (training implicit generative models)?

-What exactly is the use of this generalization bound? What does it tell us? What are the actual situations in which it holds? Is it possible that it will ever be relevant to training GANs or to developing new methods for training GANs?

Experiments:
-I appreciate that results are taken over 10 different random seeds.

-If the choice of gamma is unimportant then why is it different for one experiment? I found footnote 4 confusing and contradictory.

-For figure 3, I do not think that the margin is "significant"--it constitutes a relative 6.5% improvement, which I do not believe really justifies the increased complexity and compute cost of the method.

-I appreciate Table 1 and Figure 4 for elucidating (a) how orthogonal the U and V matrices end up and (b) the observed decay of the spectra.

Appendix:
-Please change table 7 to be more readable, with captions underneath each figure rather than listed at the top and forcing readers to count the rows and match them to the caption. What is the difference between SN-GAN and Spectral Norm in this table? Or is that a typo, and it should be spectral-constraint?

-I Would like to see a discussion of table 7 / interpretation of why the spectra look that way (and why they evolve that way over training) for each regularizer.

Minor:
-Typos and grammatical mistakes throughout.
-As per the CIFAR-10/100 website (https://www.cs.toronto.edu/~kriz/cifar.html) the Torralba citation is not the proper one for the CIFAR datasets, despite several recent papers which have used it.
-Intro, last paragraph, "Generation bound" should be generalization bound?
-Page 4, paragraph 2, last sentence, problem is misspelled.

---

> ### Author Response · Authors · 2018-11-26
> **Response to Reviewer 2**
>
> Thank you for your valuable comments. We correct typos and grammatic mistakes in the revised version. In the following, we summarize your comments and our responses. Please also refer to the revised version for more details.
>
> Comments: Figure 1 is confusing.
>
> Response:
> Please see the updated Figure 1 in our revised version. We compare the singular value decay patterns of different methods. Note that not all layers decay slowly in SN-GAN. For a better demonstration, we label the slow/fast singular value decay on these patterns.
>
> Comments: The motivation of D-Optimal regularizer is unclear.
>
> Response:
> A detailed explanation of D-Optimal design has been added on Page 5 of the revised version. In short, a slow decay is essentially encouraging the network to capture as many features as possible while allowing neurons to be correlated. The D-Optimal regularizer is motivated by D-optimal design. The D-optimal design is a popular principle in experimental design, where people aim to estimate the parameters of statistical models with a minimum number of experiments. Specifically, D-optimal design maximizes the determinant of Fisher information matrix while allowing correlation between features in experiments. Analogously, our proposed D-Optimal Regularizer essentially maximizes the log Gram determinant of the weight matrix.
>
> Comments: The explanation of Divergence Regularizer is unclear.
>
> Response:
> The idea of divergence regularizer is to push singular values toward a desired slow decay curve. We introduce the Divergence Regularizer mainly for a comparison among different regularizers. Imposing a polynomial logarithmic function as a reference is very interesting. We haven't tried it, and will add it in the next revision.
>
> Comments: Subjective Language.
>
> Response:
> In the revised version, we use more objective language when describing our contributions. For example, in the last paragraph of section 2, we use "...both combinations improve the training of GANs on the CIFAR-10 and STL-10 datasets".
>
> Comments: Questions related to Generalization (Section 3)
>
> Response:
> A small generalization bound for GANs implies that the generated distribution is close to the true data distribution if the training error is also small. Theorem 2 justifies the benefit of controlling spectral norms in GANs, as SN-GAN and our result show that normalizing the largest singular value yields better performance than the original DC-GAN. On the other hand, as discussed in Remark 4, we are still lacking tools to characterize the effect of slow singular value decay in generalization, and we leave it for future investigation.
>
> Comments: Hyperparameters setting is unclear.
>
> Response:
> The regularization parameter, gamma, is chosen according to the output range of different regularizers. We set a smaller gamma for Divergence Regularizer, since its output is much larger than other regularizers. We make a clarification in the revised version.
>
> As mentioned in the revised footnote 4, when lambda in [1,100] and gamma in [0.2,5] (gamma in [0.01,0.1] for Divergence Regularizer), the difference in performance of GANs is negligible. However, using extremely large/small hyperparameters does undermine the performance.
>
> Comments: Table 7 is confusing.
>
> Response:
> We make the following changes in Table 7 accordingly. We add captions and clarify the difference between the spectral normalization with power iteration and with SVD reparamerterization. We also explain why the spectrum distributes in a certain way for each regularizer in Appendix D.3. See more details in the Appendix of the revised version.

---

### Official Review · AnonReviewer1 · 2018-11-01
**Improves stability of training of GANs**

**Rating:** 6
**Confidence:** 2

**Review:**

The paper builds on the experimental observations made in Miyato et al. (2018) in which the authors highlight the utility of spectral normalization of weight matrices in the discriminator of a GAN to improve the stability of the training process. The paper proposes to reparameterize the weight matrices by something that looks like the singular value decomposition, i.e. W = U E V^T. Four different techniques to control the spectrum of W by imposing various constraints on E have been discussed. For maintaining the orthonormality of U and V penalties are added to the cost function. The paper also derives a bound on the generalization error and experimentally shows the "desirable slow decay"  of singular values in weight matrices of the discriminator. Other experiments which compare the proposed approach with the SN-GAN have also been given.

(1)The paper puts a lot of stress on the stability of the training process in the beginning but clear experiments supporting their claim related to improved "stability" are lacking.
(2)It would be helpful for the readers if more clarity is added to the paper with respect to the desirability of "slow decay of singular values" and spectral normalization.
(3)The point regarding convolutional layers should be part of the main paper.

---

> ### Author Response · Authors · 2018-11-26
> **Response to Reviewer 1**
>
> Thank you for your valuable comments. We summarize our responses as follows and please refer to the revised version for more details.
>
> (1) We briefly mention stability issue in the introduction (we have revised the first sentence of the abstract). The stability issue exists generally for training GANs, which stems from solving for equilibria of the nonconvex-nonconcave min-max problem (see the second paragraph on page 2 of the revised version). Empirical success demonstrates that imposing regularizations, such as spectral normalization and gradient penalty, can ease such a stability issue. This motivates our proposed methodology to manipulate on the spectrum of the weight matrix.
>
> (2) We make a clarification in Section 2.2.2 of the revised version on the idea of ``slow decay of singular values'' by comparing orthogonal regularization (no decay), spectral normalization under power iteration (slow decay), and spectral normalization with SVD (fast decay). We also highlight that using power iteration for spectral normalization encourages a slow singular value decay, in contrast to the fast decay yielded by standard spectral normalization.
>
> (3) We explain the detailed implementation of convolutional layers in the second paragraph of Section 4 in the revised version.

---

### Official Review · AnonReviewer3 · 2018-11-02
**Interesting paper that proposes a way to control the spectrum of the networks weights**

**Rating:** 8
**Confidence:** 4

**Review:**

The paper is a natural extension of [1] which shows the importance of spectral normalization to encourage diversity of the discriminator weights in a GAN. A simple and effective parametrization of the weights similar to SVD is used: W = USV^T is used along with an orthonormal penalty on U and V and spectral penalty to control the decay of the spectrum. Unlike other parametrizations of orthogonal matrices which are exact but computationally expensive, the proposed one tends to be very accurate in practice and much faster.  A generalization bound is provided that shows the benefit of controlling the spectral norm. Experimental results show that the method is accurate in constraining the orthonormality of U and V and in controlling the spectrum. The experiments also show a marginal improvement of the proposed method over SN-GAN [1].
However, the following it is unclear why one would want to control the whole spectrum when theorem 2 only involves the spectral norm. In [1], it is argued that this encourages diversity in the weights which seems intuitive. However, it seems enough to use Spectral Normalization to achieve such purpose empirically according to that same paper. It would be perhaps good to have an example where SN fails to control the spectrum in a way that significantly impacts the performance of the algorithm while the proposed method doesn't.

Overall the paper is clearly written and the proposed algorithm effectively controls the spectrum as shown experimentally, however,  given that the idea is rather simple, it is important to show its significance with examples that clearly emphasize the importance of controlling the whole spectrum versus the spectral norm only.


Revision: Figure 1 is convincing and hints to why SN-GAN acheives slow decay while in principle it only tries to control the spectral norm. I think this paper is a good contribution as it provides a simple and efficient algorithm to precisely control the spectrum. Moreover, a recent work ([2], theorem 1 ) provides theoretical evidence for the importance of controling the whole spectrum which makes this contribution even more relevant.


[1] T. Miyato, T. Kataoka, M. Koyama, and Y. Yoshida. Spectral Normalization for Generative Adversarial Networks. Feb. 2018.
[2] M. Arbel, D. J. Sutherland, M. Bin ́kowski, and A. Gretton. On gradient regularizers for MMD GANs. NIPS 2018

---

> ### Author Response · Authors · 2018-11-26
> **Response to Reviewer 3**
>
> Thank you for your valuable comments. We discuss the raised questions as follows.
>
> [1] owe their empirical success of training SN-GAN to controlling the spectral norm while allowing flexibility. This perspective, however, is not very concrete. As we know, orthogonal regularization and spectral normalization with SVD can both control the spectral norm. Their empirical performance is actually worse than SN-GAN. For example, on the STL-10 dataset, SN-GAN achieves an inception score of 8.83, while spectral normalization with SVD only achieves 8.69 and orthogonal regularization achieves 8.77. The reason behind is that SN-GAN implements the spectral normalization via one-step power iteration. This procedure consistently underestimates spectral norms of weight matrices. Consequently, in addition to controlling the spectral norms, the spectral normalization in SN-GAN affects the whole spectrum of the weight matrix (encourages slow singular value decay as in Figure 1), which we refer to as ``flexibility''. Built upon these empirical observations, we conjecture that controlling the whole spectrum better improves the performance of GANs, which is further corroborated by our numerical experiments. This discussion has been added to the beginning of Section 2.2.2 in the revised version.
>
> Theorem 2 justifies the benefit of controlling spectral norms in GANs. [1] and our result both show that normalizing the largest singular value yields better performance than the original DC-GAN. On the other hand, as discussed in Remark 4, we are still lacking tools to characterize the effect of slow singular value decay in generalization as well as preventing mode collapse. Thus, we leave it for future investigation.
>
> [1] T. Miyato, T. Kataoka, M. Koyama, and Y. Yoshida. Spectral Normalization for Generative Adversarial Networks. Feb. 2018.

---

### Meta-Review · Area_Chair1 · 2018-12-17

**Confidence:** 3
**Recommendation:** Accept (Poster)

**Metareview:**

All the reviewers agree that the paper has an interesting idea on regularizing the spectral norm of the weight matrices in GANs, and a generalization bound has been shown. The empirical result shows that indeed regularization improves the performance of the GANs. Based on these the AC suggested acceptance.